# Leptin increases sympathetic nerve activity via induction of its own receptor in the paraventricular nucleus

**Zhigang Shi**[1†], **Nicole E Pelletier**[1†], **Jennifer Wong**[1], **Baoxin Li**[1], **Andrei D Sdrulla**[2], **Christopher J Madden**[3], **Daniel L Marks**[4*], **Virginia L Brooks**[1*]

[1]Department of Physiology and Pharmacology, Portland, United States; [2]Department of Anesthesiology and Perioperative Medicine, Portland, United States; [3]Department of Neurological Surgery, Portland, United States; [4]Department of Pediatrics, Pape Family Pediatric Research Institute, Brenden-Colson Center for Pancreatic Care Oregon Health & Science University, Portland, United States

**Abstract** Whether leptin acts in the paraventricular nucleus (PVN) to increase sympathetic nerve activity (SNA) is unclear, since PVN leptin receptors (LepR) are sparse. We show in rats that PVN leptin slowly increases SNA to muscle and brown adipose tissue, because it induces the expression of its own receptor and synergizes with local glutamatergic neurons. PVN LepR are not expressed in astroglia and rarely in microglia; instead, glutamatergic neurons express LepR, some of which project to a key presympathetic hub, the rostral ventrolateral medulla (RVLM). In PVN slices from mice expressing GCaMP6, leptin excites glutamatergic neurons. LepR are expressed mainly in thyrotropin-releasing hormone (TRH) neurons, some of which project to the RVLM. Injections of TRH into the RVLM and dorsomedial hypothalamus increase SNA, highlighting these nuclei as likely targets. We suggest that this neuropathway becomes important in obesity, in which elevated leptin maintains the hypothalamic pituitary thyroid axis, despite leptin resistance.

**\*For correspondence:**
marksd@ohsu.edu (DLM);
brooksv@ohsu.edu (VLB)

[†]These authors contributed equally to this work

**Competing interests:** The authors declare that no competing interests exist.

## Introduction

Obesity activates the sympathetic nervous system, which can lead to hypertension, in part due to the central actions of adipose-derived leptin (*Hall et al., 2010*; *Bell and Rahmouni, 2016*). Multiple sympathoexcitatory leptin-receptive sites exist within the hypothalamus (*Harlan and Rahmouni, 2013*). All of these sites appear to converge in the hypothalamic paraventricular nucleus (PVN), since PVN blockade completely reverses the increases in sympathetic nerve activity (SNA) induced by infusion of leptin into the cerebroventricles (icv) (*Shi et al., 2015*). However, combined blockade of major excitatory neuronal inputs into the PVN only partially reverses the increases in sympathetic nerve activity (SNA) (*Shi et al., 2015*), suggesting that other excitatory mechanisms in PVN contribute.

One possible mechanism is that leptin also acts directly in the PVN to increase SNA. However, PVN leptin receptor (LepR) expression in the PVN is scarce (*Guan et al., 1997*; *Elmquist et al., 1998*; *Scott et al., 2009*). Moreover, previous studies failed to detect a sympathoexcitatory effect of PVN leptin (*Marsh et al., 2003*; *Montanaro et al., 2005*), although the observational periods were short, less than 30 min. Since icv leptin injection produces a slowly developing increase in SNA (*Shi et al., 2015*), and since leptin can induce LepR expression, at least in cultured microglia (*Tang et al., 2007*), we hypothesized that leptin increases the expression of its own receptor in PVN, thereby facilitating direct or indirect activation of presympathetic neurons. To test this hypothesis, we first determined if PVN nanoinjections of leptin increase SNA in α-chloralose anesthetized rats. A key aspect of our approach was that we monitored lumbar SNA (LSNA) for 2 hr after the injection.

Second, we employed a recently developed fluorescent in situ hybridization (FISH) approach that produces robust amplification of the mRNA signal, to establish if and where LepR are expressed in the PVN. Third, we quantified LepR expression after PVN leptin nanoinjections using classical ISH.

The cellular-molecular mechanisms by which leptin increases SNA are relatively unexplored in any brain region, but in particular in the PVN. Infusion of leptin icv increases arterial pressure (AP) and SNA via activation of receptors that bind glutamate or the proopiomelanocortin (POMC) product, α-melanocyte stimulating hormone (α-MSH) (*Yu and Cai, 2017*), specifically in the PVN (*Shi et al., 2015*). Moreover, in other brain regions, leptin can increase ionotropic glutamate receptor trafficking (*Moult and Harvey, 2009*) and inhibit astroglial glutamate uptake (*Fuente-Martín et al., 2012*), thereby enhancing glutamatergic excitation. Therefore, we next tested if PVN leptin increases SNA in part via local activation of melanocortin type 3/4 receptors (MC3/4R) or glutamatergic receptors (GluR).

We next sought to identify the cells that express the LepR in PVN and how these cells communicate with PVN presympathetic neurons. Microglia are a plausible candidate, since leptin's anorexic effect may involve microglia activation (*Luheshi et al., 1999*; *Pinteaux et al., 2007*), microglia can express the LepR (*Tang et al., 2007*; *Gao et al., 2014*; *Lafrance et al., 2010*), and leptin can activate microglia in vitro (*Lafrance et al., 2010*). Moreover, leptin and microglia were recently identified as key players in hypertension-provoking influences of obesity (*Xue et al., 2016*), although the specific brain sites and cell types involved in this interaction are unknown. On the other hand, *lepr* mRNA has also been detected in astroglia (*Kim et al., 2014*; *Hsuchou et al., 2009*), and leptin's ability to suppress feeding also may involve astroglia (*Kim et al., 2014*). Finally, LepR may be expressed in PVN presympathetic neurons or local interneurons that target these cells. Therefore, using specific combinations of ISH, FISH, and immunohistochemistry (ihc), we interrogated the presence of LepR in microglia, astroglia, and presympathetic neurons, which were identified by injecting the retrograde tracer, cholera toxin B (CTB), into the rostral ventrolateral medulla (RVLM).

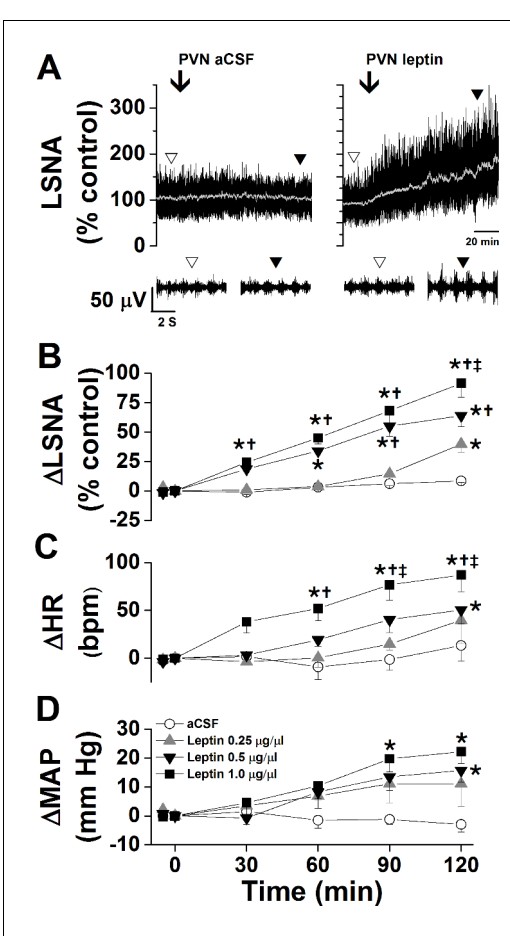

**Figure 1.** Effects of PVN leptin on LSNA, HR, and MAP. (**A**) Representative experiments showing that bilateral nanoinjections (beginning at arrow) of leptin, but not aCSF, into the PVN increased LSNA. (**B-D**) Bilateral nanoinjections of leptin, but not aCSF, into the PVN dose dependently increased LSNA (**B**), HR (**C**), and MAP (**D**). Open circles, PVN aCSF (n = 4); gray triangles, PVN leptin, 15 ng (n = 4); black triangles, PVN leptin, 30 ng (n = 4); black squares, PVN leptin, 60 ng (n = 5). Baseline HR values (in bpm; no between group difference) are: PVN aCSF: 354 ± 12; PVN leptin, 15 ng: 369 ± 7; PVN leptin, 30 ng: 331 ± 12; PVN leptin, 60 ng: 311 ± 18. Baseline MAP values (in mmHg; no between group differences) are: PVN aCSF: 103 ± 7; PVN leptin, 15 ng: 103 ± 9; PVN leptin, 30 ng: 106 ± 5; PVN leptin, 60 ng: 98 ± 3. *: p<0.05, compared to baseline control values (time zero); †: p<0.05 compared to PVN leptin, 15 ng; ‡: p<0.05 compared to PVN leptin, 30 ng.

## Results

### PVN leptin dose-dependently and specifically increases LSNA, MAP, and HR and enhances baroreflex control of LSNA and HR

Male Sprague-Dawley rats were anesthetized and prepared for hypothalamic nanoinjections, and for measurements of mean AP (MAP), heart rate (HR), LSNA, and baroreceptor reflex control of LSNA and HR, as previously described

(*Li et al., 2013*). Briefly, to assess baroreflex function, we slowly lowered MAP with increasing iv doses of the vasodilator, nitroprusside, and then raised MAP by reducing the nitroprusside infusion rate and by increasing an iv infusion of the vasoconstrictor, phenylephrine. The resulting changes in LSNA/HR were related to the changing MAP, and a sigmoidal baroreflex curve was fitted to the data. The fit allowed determination of the maximum and minimum LSNA/HR at low and high MAP, respectively, as well as an assessment of baroreflex sensitivity or gain, which is the maximum slope of the most linear part of the curve. Our rationale for including baroreflex measurements was two-fold: 1) icv leptin enhances baroreflex function by increasing baroreflex gain and the maximum reflex-induced increase in SNA at low MAP (*Shi et al., 2015*; *Li et al., 2013*); however, whether PVN leptin similarly enhances the baroreflex is unknown. Food (carbohydrate) consumption increases leptin levels (*Havel, 2004*) and decreases MAP, which could result in exaggerated baroreflex increases in SNA. Importantly, eating is a well-established trigger for myocardial infarction, due to increased SNA (*Culic, 2007*; *Nawrot et al., 2011*), suggesting that the interplay between leptin and the baroreflex is significant. 2) Subtle differences in the effect of PVN leptin on baroreflex function between icv infusions and various hypothalamic injections may serve to establish specificity of the actions of PVN leptin.

As shown in representative experiments in *Figure 1A* and grouped data in *Figure 1B–D*, PVN leptin dose dependently increased LSNA; only the two higher doses significantly increased MAP and HR. The increases were all slowly developing, taking 30–120 min to reach significant levels. On the other hand, PVN aCSF was without effect. PVN leptin also enhanced baroreflex control of LSNA and HR, by increasing the baroreflex gain and maximum (*Figure 2A–D*). PVN injections of aCSF did not alter baroreflex control of LSNA/HR (*Figure 2E–F*). Interestingly, these effects on baroreflex control of LSNA are different from the actions of icv leptin, in which the LSNA baroreflex minimum was also increased (*Li et al., 2013*). In contrast, injections of leptin outside of the PVN (n = 4) failed to significantly alter LSNA (101 ± 1 to 103 ± 13%), MAP (112 ± 6 to 117 ± 6 mmHg), or HR (340 ± 13 to 342 ± 7 bpm) within 2 hr.

As another anatomical control, we injected leptin into the nearby DMH. While DMH injection of leptin also slowly increased LSNA, neither MAP nor HR was significantly altered (*Figure 3A*). DMH leptin also enhanced baroreflex control of LSNA by increasing baroreflex gain, the maximum, and notably, the minimum (*Figure 3A*), similarly to the effects of blockade of DMH GABA$_A$ receptors (*McDowall et al., 2006*). In sharp contrast to the effects of icv or PVN leptin, DMH leptin increased HR baroreflex gain, but no other HR baroreflex parameters (*Figure 3A*). Nanoinjections of leptin into the arcuate nucleus (ArcN) also increased LSNA and its baroreflex regulation (gain), but while HR was increased, baroreflex control of HR was unaltered (*Figure 3B*). These data indicate that leptin acts specifically in the PVN (DMH and ArcN) to increase LSNA and its baroreflex regulation.

Leptin acts in several hypothalamic sites to increase SNA to brown adipose tissue (BAT). While the PVN has not been studied, a recent report (*Cakir et al., 2019*) revealed that deletion of the LepR in Sim1 (PVN) neurons decreased BAT temperature at ambient temperature and during cold stress, the latter of which was associated with a larger fall in thyroxine levels. The fall in thyroxine alone can explain the reduced BAT temperature; however, in parallel PVN leptin may also activate BAT SNA. In support, we found that bilateral nanoinjection of leptin into the PVN of rats with normal body temperature (~37°C) increased BAT SNA (*Figure 4A*). BAT temperature increased in 4 of 5 experiments, but overall BAT temperature was not significantly altered (*Figure 4A*). Because cooling can amplify BAT responses (*Tupone et al., 2011*), we next tested leptin in rats held at ~35°C. In this case, PVN leptin significantly increased both BAT SNA and BAT temperature (*Figure 4B*).

We previously reported that icv leptin increases LSNA in females only during proestrus, because this action requires elevated levels of estrogen (*Shi and Brooks, 2015*). As with icv leptin, PVN leptin was ineffective in ovariectomized (OVX) females, but significantly increased LSNA in OVX females in which proestrus levels of estrogen were replaced with a silastic implant (*Figure 5*).

Collectively, these data indicate that leptin acts in the PVN to increase SNA to muscle and BAT, in males as well as females in the presence of proestrus levels of estrogen.

## LepR expression in the PVN

While the distribution of LepR expression in the PVN has been deduced indirectly via quantification of leptin-induced p-STAT3 expression (*Perello et al., 2006*; *Huo et al., 2004*), no previous studies detailed PVN LepR localization. Detection of PVN LepR using classical ISH is challenging

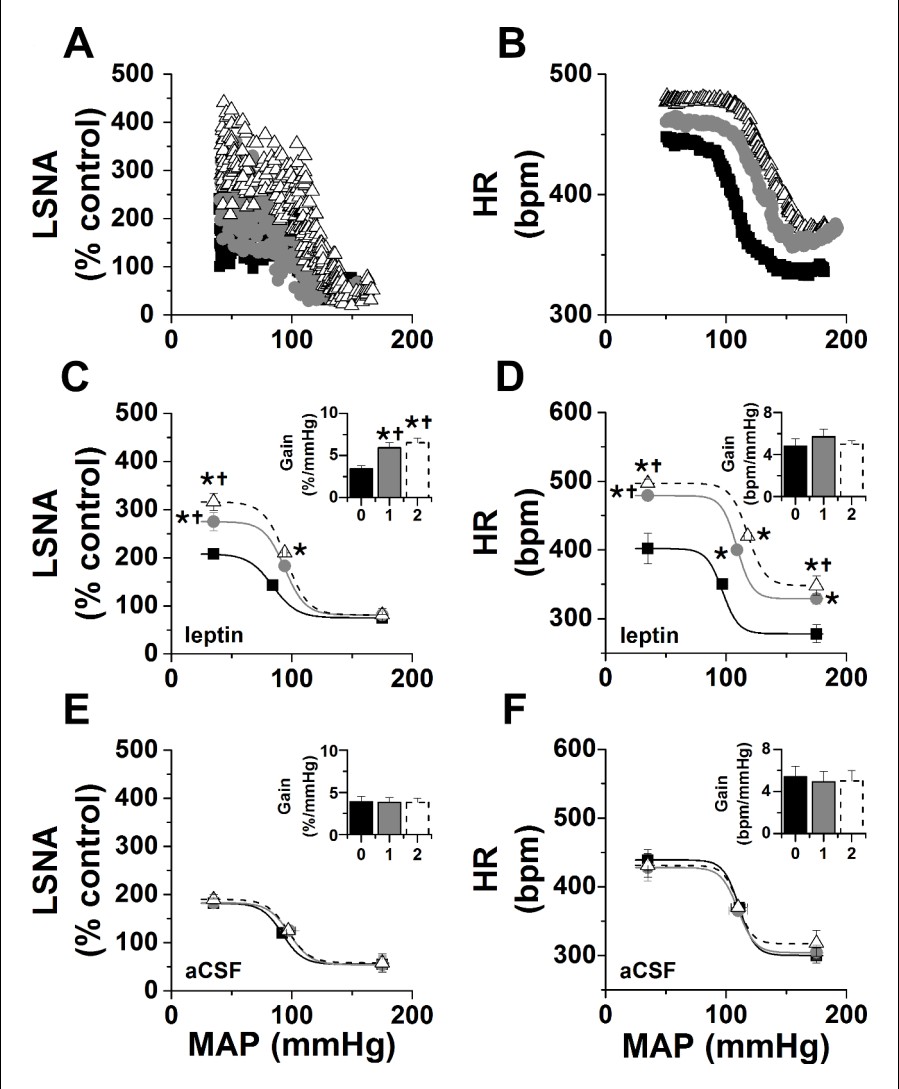

**Figure 2.** Effects of PVN leptin on baroreflex control of LSNA and HR. (A-B) Representative experiments showing effects of PVN leptin (60 ng) on baroreflex control of LSNA (A) and HR (B). (C) Grouped LSNA data. PVN nanoinjection of leptin enhanced baroreflex control of LSNA (increased baroreflex gain, maximum, BP50). (D) Grouped HR data. PVN leptin also increased baroreflex control of HR (increased baroreflex maximum, minimum). Closed black squares, black bars, solid line: baseline; closed gray circles, gray bar, solid gray line: 1 hr after PVN leptin or aCSF; open black triangles, open dashed bar, dashed line: 2 hr after PVN leptin or aCSF. *: p<0.05, compared to baseline; †: p<0.05 compared to icv aCSF at the same time.

(*Guan et al., 1997*; *Elmquist et al., 1998*), given its weak signal. Therefore, we used RNAScope, which includes a proprietary amplification step of the mRNA signal, to survey LepR in the PVN in rats (*Figure 6*). LepR expression was higher in the rostral and caudal PVN, with lower signal observed in between. In all PVN regions, the signal intensity was less than in the ArcN, a well-established hypothalamic target of leptin. In mice (*Figure 7*), as in rats, LepR expression was detectable, but again the level was far less than in the ArcN.

## PVN leptin induces the expression of the LepR

We next tested whether leptin's slowly developing sympathoexcitatory action was due in part to its ability to increase the expression of its own receptor. We quantified LepR signal intensity on a side of the PVN injected with leptin and compared it to the opposite aCSF-injected side. A single unilateral injection of leptin (60 ng) into the PVN produced a delayed (by ~2 hr) but sustained (~2–4 hr)

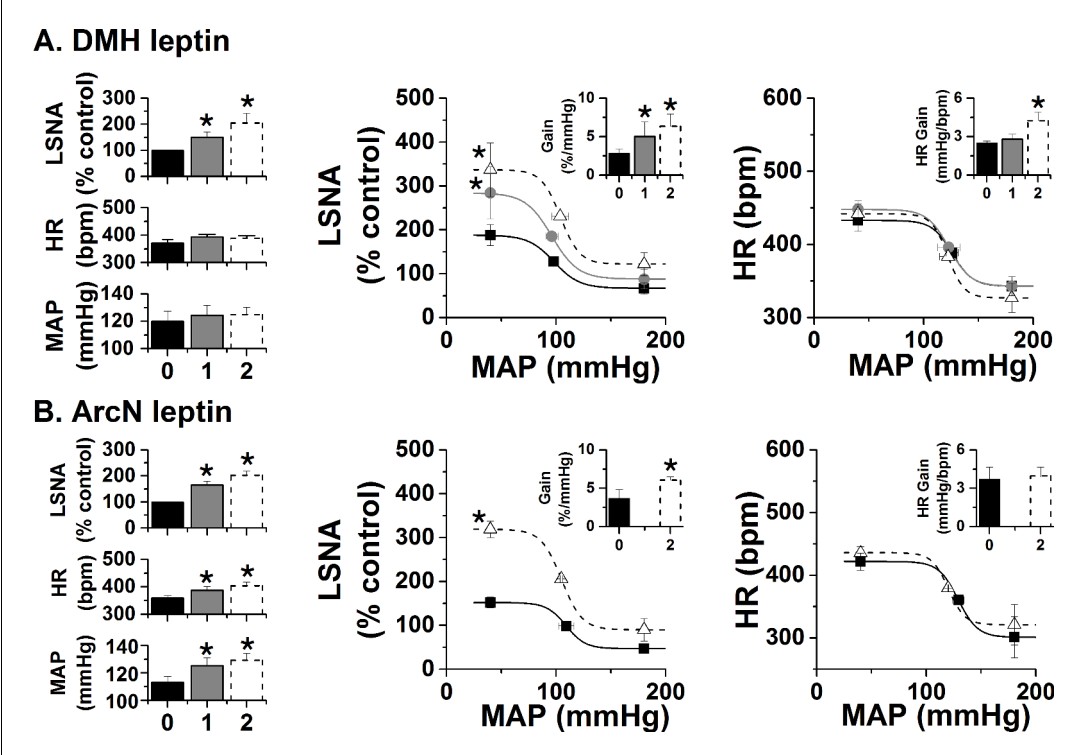

**Figure 3.** Effects of DMH and ArcN leptin on LSNA, HR, and MAP, and baroreflex control of LSNA and HR. (**A**) Bilateral DMH nanoinjections of leptin (n = 4) enhanced basal LSNA and baroreflex control of LSNA (increased baroreflex gain, maximum, minimum); however, while HR gain was enhanced, basal HR and MAP and other HR baroreflex parameters were not significantly altered. *: $p < 0.05$, compared to baseline control values. (**B**) Bilateral ArcN nanoinjections of leptin (n = 6) enhanced basal LSNA and baroreflex control of LSNA (increased baroreflex gain and maximum), HR and MAP; however, baroreflex control of HR was unaltered. Closed black bars, solid black squares, solid line: baseline; closed gray bars, closed gray circles, solid gray line: 1 hr after DMH/ArcN leptin; open dashed bar, open black triangle, dashed line: 2 hr after DMH/ArcN leptin. *: $p < 0.05$, compared to baseline control values.

increase in LSNA, HR, and MAP. Four to six hr after leptin injection (when the responses were stable), brains were collected, snap-frozen and processed for ISH for the LepR (white) and oxytocin [red; used to define the PVN region-of-interest (ROI)], as shown in *Figure 8*. As previously reported in otherwise untreated rats (*Guan et al., 1997*; *Elmquist et al., 1998*), PVN LepR signal detected using ISH was very low in aCSF-treated sides of the PVN. Parallel to observations made with FISH, this expression of LepR, while detectable, was far less than that in the neighboring ArcN. More importantly, PVN leptin nanoinjections induced significant elevations in LepR expression, compared to contralateral aCSF injected side (*Figure 8*). The LepR induction was observed only in the more caudal levels of PVN. LepR were rarely detected in OT neurons, although tended to be more common on the leptin (2.9 ± 0.6% of OT neurons) versus the aCSF sides (0.7 ± 0.1% of OT neurons). Thus, leptin can increase the expression of its own receptor in vivo, as previously shown in vitro.

## Blockade of PVN ionotropic glutamate receptors (iGluR), but not of melanocortin type 3 or 4 receptors (MC3/4R), partially reversed PVN leptin-induced increases in LSNA, MAP, and HR

Because even after leptin injection, LepR expression was low, we next investigated if leptin synergizes with excitatory inputs into the PVN. *Figure 9* summarizes experiments designed to investigate the local participation of glutamate and α-MSH in the sympathoexcitatory response to PVN leptin. As in *Figure 1*, ninety min after bilateral injections of leptin into the PVN, LSNA, MAP and HR were all significantly elevated (*Figure 9*). Subsequent bilateral PVN injection of kynurenic acid (KYN) to block iGluR promptly decreased these variables; however, LSNA remained elevated compared to baseline (*Figure 9A–D*). In contrast, PVN injections of SHU9119 had no effects (*Figure 9E–F*).

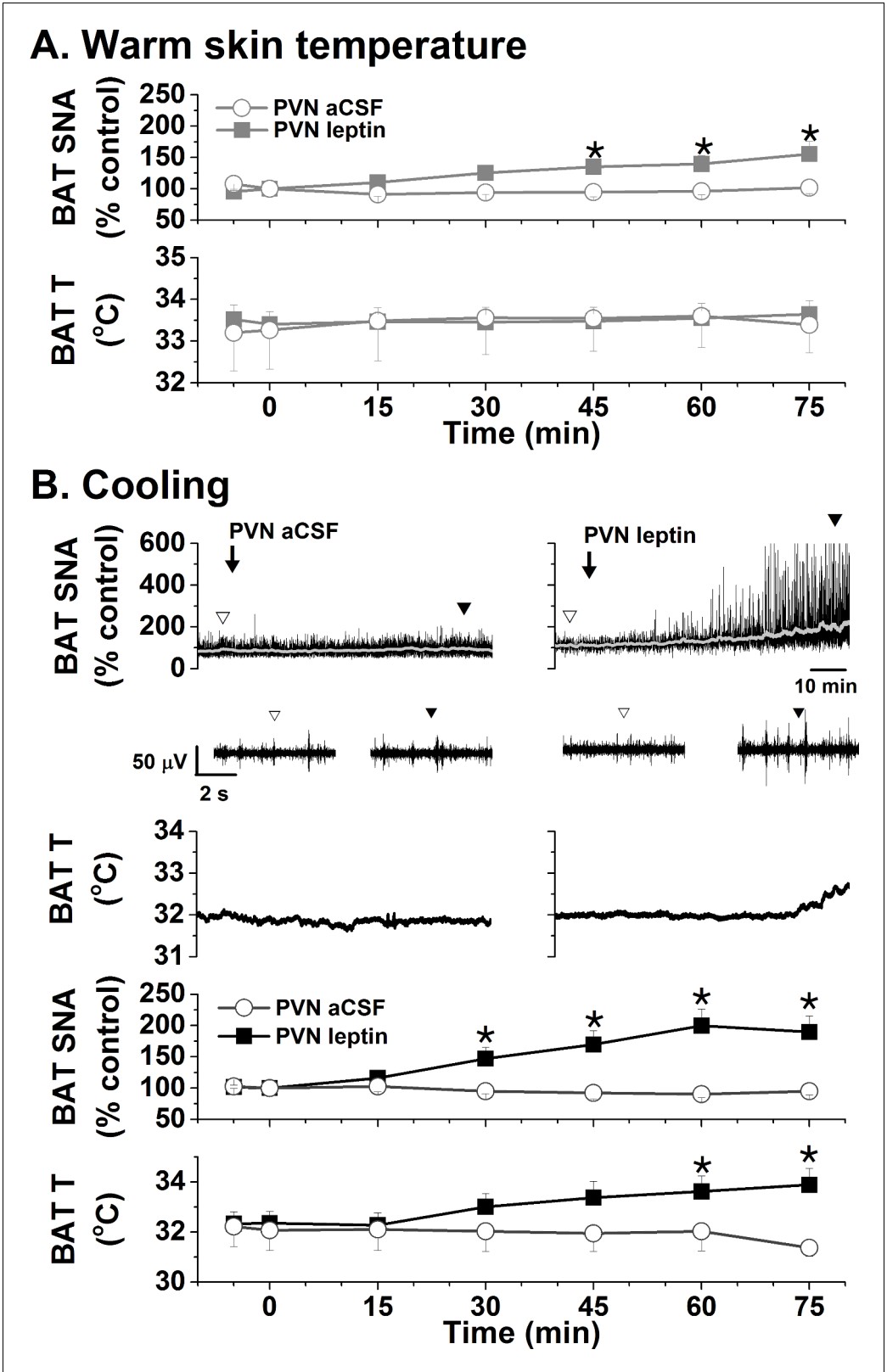

**Figure 4.** Effects of PVN leptin on BAT SNA in warm and cooled rats. (**A**) In rats with warm skin temperature (37.2 ± 0.2°C), PVN leptin (n = 7) increased BAT SNA, without significantly altering BAT temperature. PVN aCSF (n = 4) did not alter these variables. (**B**) Top, representative experiments; bottom, grouped data. In cooled rats (35.4 ± 0.1°
*Figure 4 continued on next page*

*Figure 4 continued*

C skin temperature), PVN leptin (n = 6) increased BAT SNA, and BAT temperature. Again, PVN aCSF (n = 4) was without effect. *: p<0.05, compared to baseline control values.

Moreover, neither KYN nor SHU9119 altered LSNA, MAP or HR following PVN injections of aCSF (*Figure 9*, open bars). Thus, local PVN glutamate, acting via ionotropic receptors, but not rapidly reversible α-MSH actions, contribute to the sympathoexcitatory effects of PVN leptin.

## LepR are rarely expressed in PVN microglia

To explicate the mechanisms by which glutamate participates in the PVN leptin-induced sympathoexcitation, we first determined if LepR are present in microglia or astroglia. Microglia, at least in culture, can express LepR (*Tang et al., 2007*; *Pinteaux et al., 2007*; *Gao et al., 2014*; *Lafrance et al., 2010*). Moreover, leptin activates cultured microglia (*Lafrance et al., 2010*) and induces and amplifies the release of proinflammatory cytokines, including IL-1β, TNF-α, and IL-6 (10,

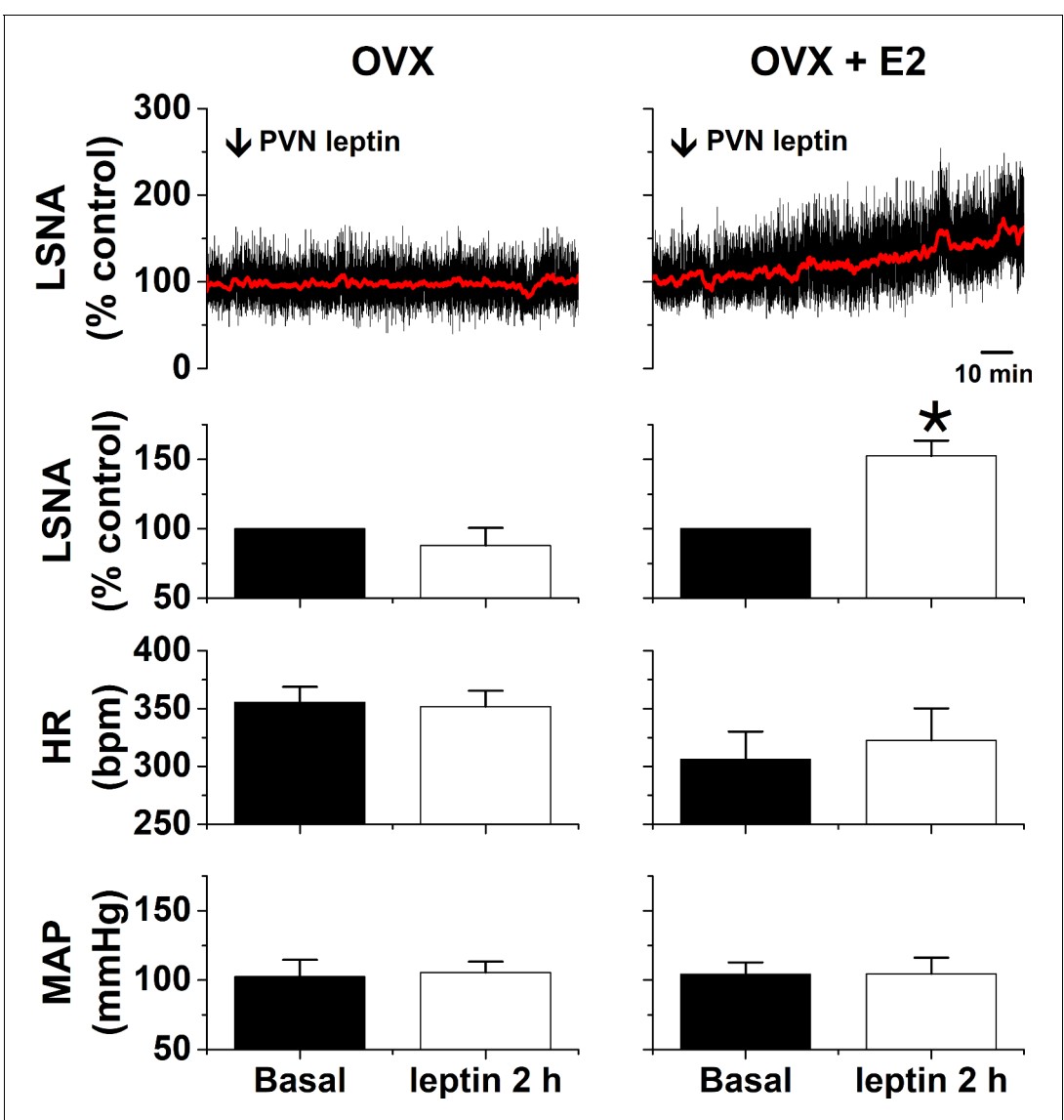

**Figure 5.** PVN leptin increases LSNA in estrogen (E2)-treated ovariectomized (OVX) female rats (n = 3), but not OVX rats (n = 3). *: p<0.05, compared to baseline control values.

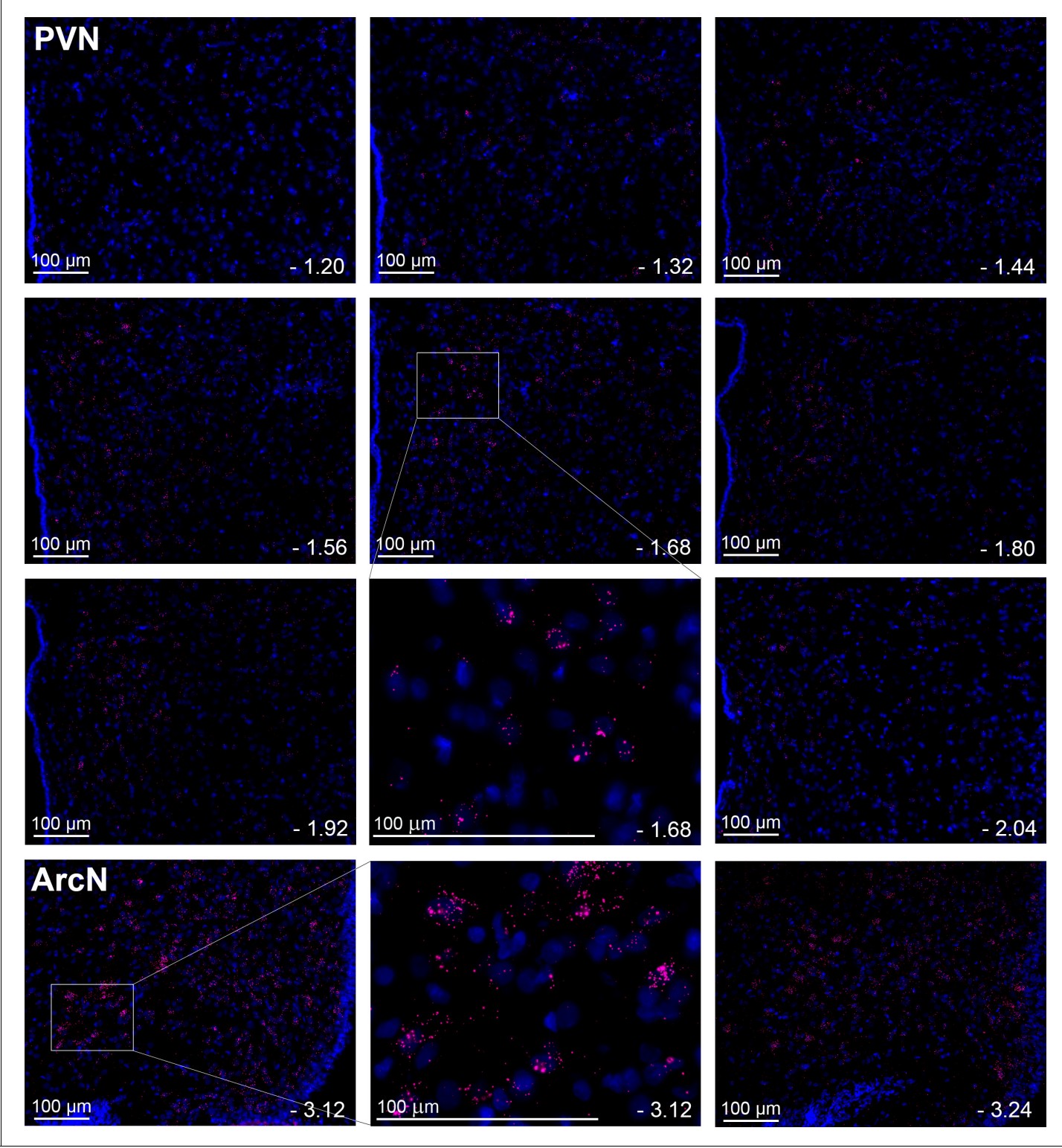

**Figure 6.** Leptin receptors (LepR) are expressed throughout the PVN in male rats, but at much lower levels than in the ArcN. Images from a representative rat showing the presence and distribution of LepR in male rats (red puncta) throughout the PVN and ArcN. Individual plates illustrate the changes in LepR distribution from rostral to caudal. PVN (representative from n = 20) and ArcN (from n = 2) levels from bregma are identified in the lower right corner of each micrograph. Higher magnification images are also shown for the PVN (−1.68 mm from bregma) and ArcN (−3.12 from bregma). Given the low LepR expression level in the PVN, red puncta are viewed optimally by using the zoom feature to survey the digital image online.

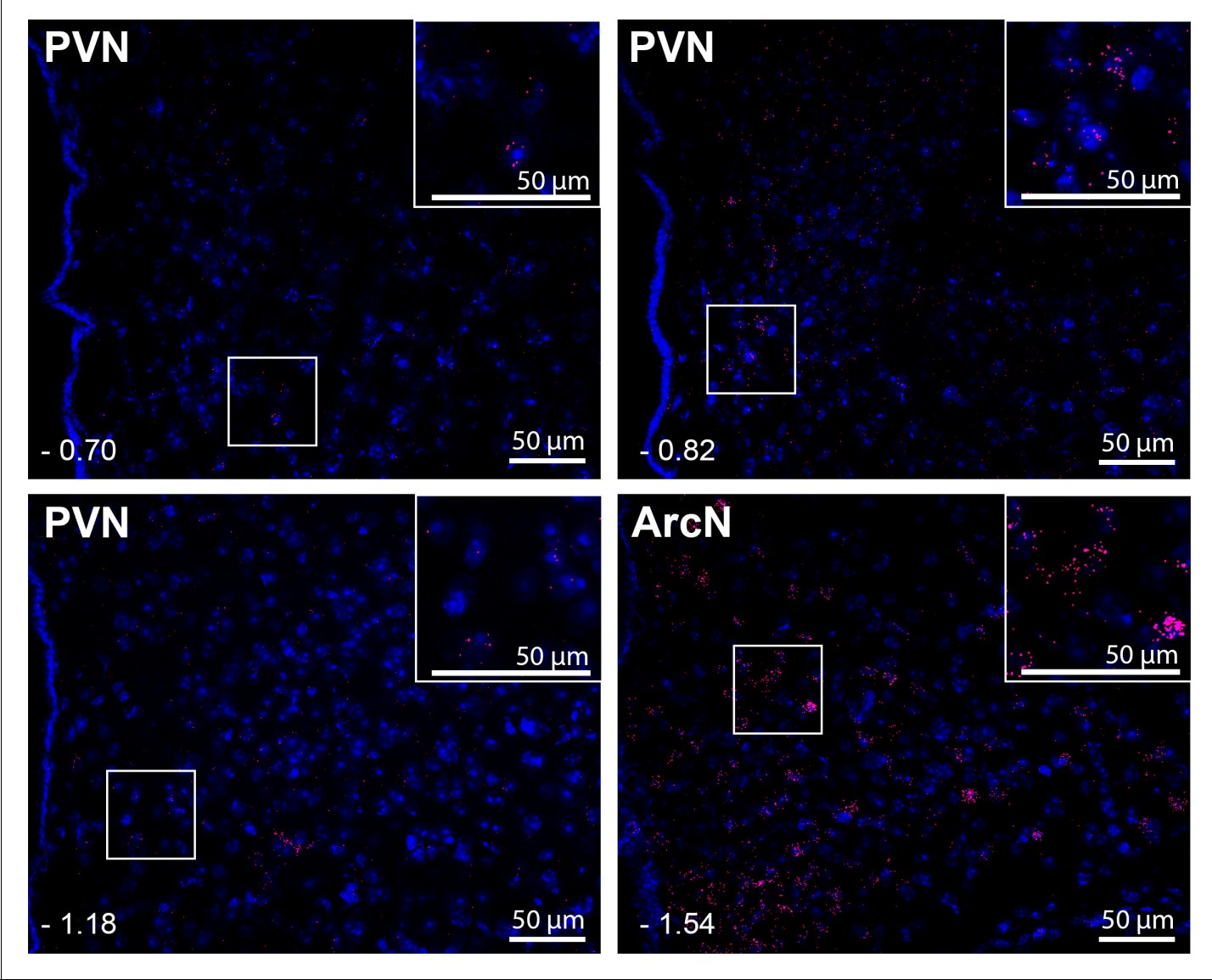

**Figure 7.** Leptin receptors (LepR) are expressed throughout the PVN in male mice, but at much lower levels than in the ArcN. Images are from one representative mouse (from n = 5). PVN and ArcN levels (mm from bregma) are displayed in the lower left corner of each micrograph. Magnified sections are also included as insets in each micrograph. As in the rat (*Figure 6*), given the low LepR expression level in the mouse PVN, red puncta are viewed best by zooming into the digital image online.

15–17). Since PVN cytokines increase SNA (*Kang et al., 2008*; *Zhang et al., 2003*; *Kang et al., 2010*; *Song et al., 2014*; *Bardgett et al., 2014a*), and cytokines like TNF-α rapidly increase the membrane expression of ionotropic glutamate receptors (*Olmos and Lladó, 2014*), leptin-induced microglial activation could indirectly increase SNA via mechanisms that involve glutamate. We first determined colocalization of iba-1 (identifies microglial cells) using ihc and LepR using FISH in untreated rats. As shown in representative images (*Figure 10A–D*), LepR were rarely detected in iba-1-immunoreactive (-ir) cells in the PVN or the ArcN–at most 1 cell per section. To ensure that all resident microglia were detected, we also identified microglia via the mRNA signal for iba-1 using RNAScope and Imaris x64v9.2.1 (Bitplane; Zurich, Switzerland) for post hoc 3D rotation of the images to confirm the colocalization (*Figure 10E–H*). Colocalization was again rare, but more evident than with the use of ihc to image iba-1 containing cells. In the PVN we observed at most 2 cells per section. In the ArcN, significantly more were observed, as previously reported (*Tang et al., 2007*; *Pinteaux et al., 2007*). Next, because leptin induces the expression of its own receptor, we

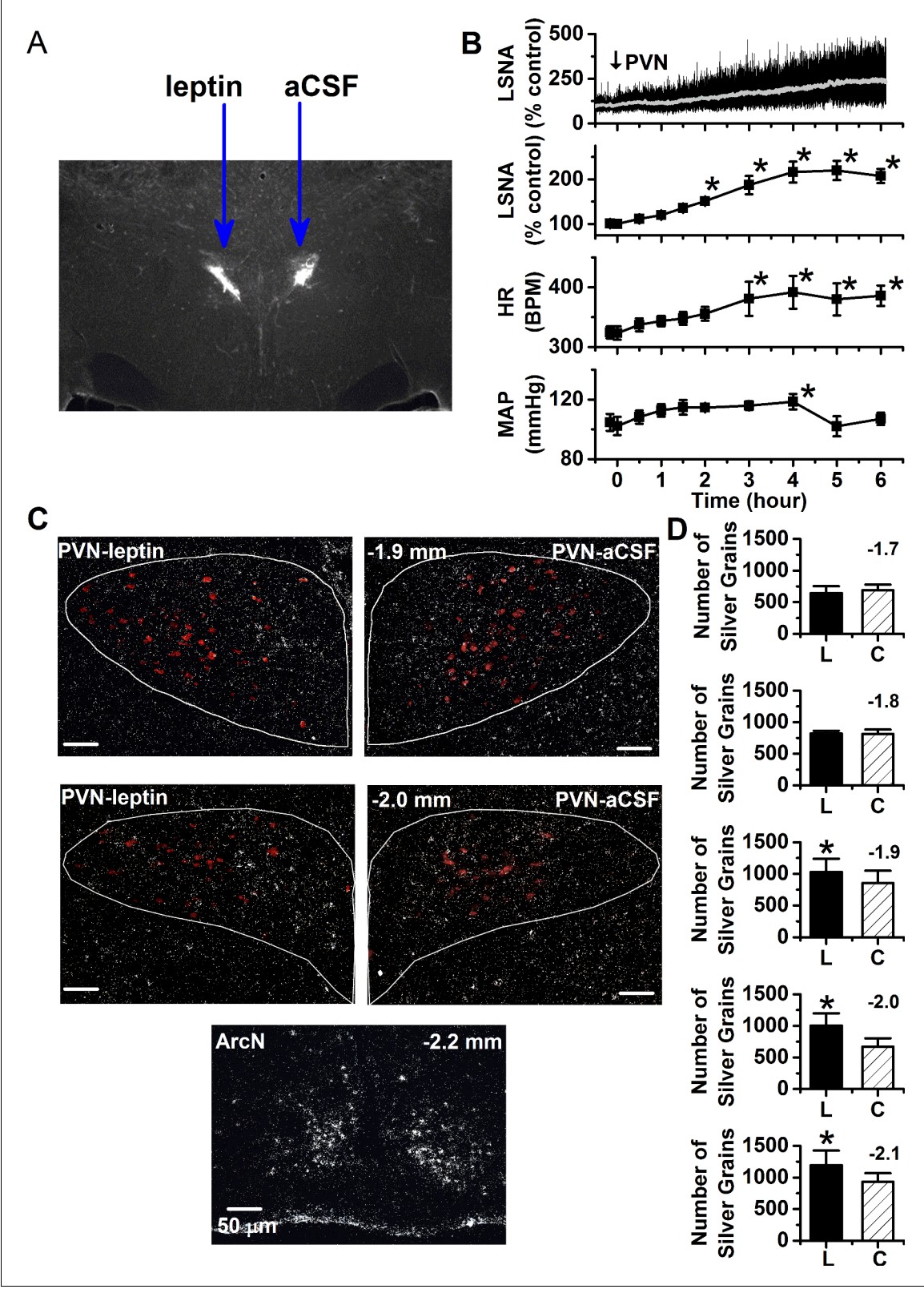

**Figure 8.** PVN leptin induces the expression of the LepR. (**A**) Representative histological image showing PVN injection sites. (**B**) Top. Representative experiment showing that a unilateral injection of leptin slowly increases LSNA. Bottom. Grouped data (n = 6) showing that a unilateral injection of leptin into the PVN slowly increases LSNA, HR, and MAP. PVN *: p<0.05, compared to baseline control values. (**C**) Top. Representative ISH images from 2 PVN levels (displayed in upper left corner of the right PVN) showing that nanoinjection of leptin (left side) increases *lepr* mRNA compared to injection

*Figure 8 continued on next page*

*Figure 8 continued*
of the aCSF vehicle (right side). Bottom. ArcN *lepr* mRNA is markedly higher than in the PVN. Oxytocin neurons in red. (D) Grouped data (n = 5) showing that PVN leptin increases LepR ISH signal compared to the contralateral aCSF-injected side. *: p<0.05, compared to contralateral aCSF injected values.

examined sections from animals (n = 3) treated with unilateral leptin and aCSF injections (*Figure 11*). Using both ihc and RNAScope for iba-1 to identify microglial cells, we again found rare overlap of the LepR and iba-1 signals in the PVN. Collectively, these data do not support a major role for microglia in the sympathoexcitation induced by PVN leptin.

## LepR are not expressed in astroglia

Several prior investigations demonstrated that astroglia also express *lepr* mRNA (*Kim et al., 2014*; *Hsuchou et al., 2009*) and that leptin alters astrocytic structure and function (*Valdearcos et al., 2015*). Astroglia tonically remove synaptically released glutamate; therefore, leptin, like other cytokines (*Tilleux and Hermans, 2007*) or angiotensin II (*Stern et al., 2016*), could directly inhibit

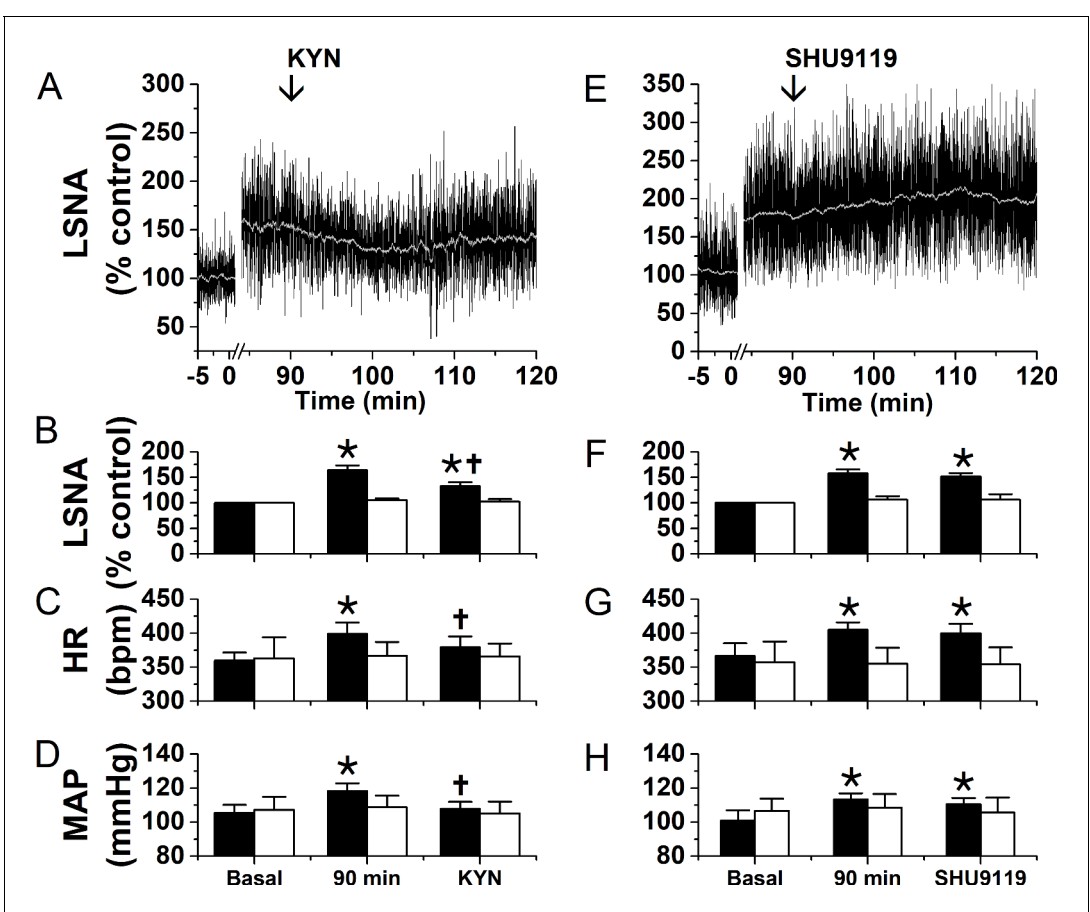

**Figure 9.** Blockade of ionotropic glutamate receptors with kynurenate (KYN), but not of MC4R with SHU9119, partially reverses the sympathexcitatory effects of PVN leptin. Left: bilateral PVN KYN injections; Right: bilateral PVN SHU9119 injections. (A) Representative experiment showing that PVN nanoinjections of leptin significantly increased LSNA after 90 min. Subsequently, bilateral PVN nanoinjections of KYN (at arrow) decreased LSNA. Grouped data show that PVN nanoinjections of leptin (black filled bars), but not aCSF (black open bars), significantly increased LSNA (B), HR (C), and MAP (D) after 90 min. Then, bilateral PVN nanoinjections of KYN (n = 6) decreased LSNA (B), HR (C), and MAP (D). PVN injections of KYN after PVN aCSF had no effects (n = 4). (E) Representative experiment showing that PVN nanoinjections of leptin significantly increased LSNA after 90 min. Subsequent bilateral nanoinjections of SHU9119 (at arrow) did not alter LSNA. (F-H) Grouped data show that bilateral PVN nanoinjections of SHU9119 had no effects in rats pretreated with either leptin (black filled bars; n = 4) or aCSF (black open bars; n = 4).*, p<0.05, compared to basal. †, p<0.05, compared with values 90 min after PVN leptin.

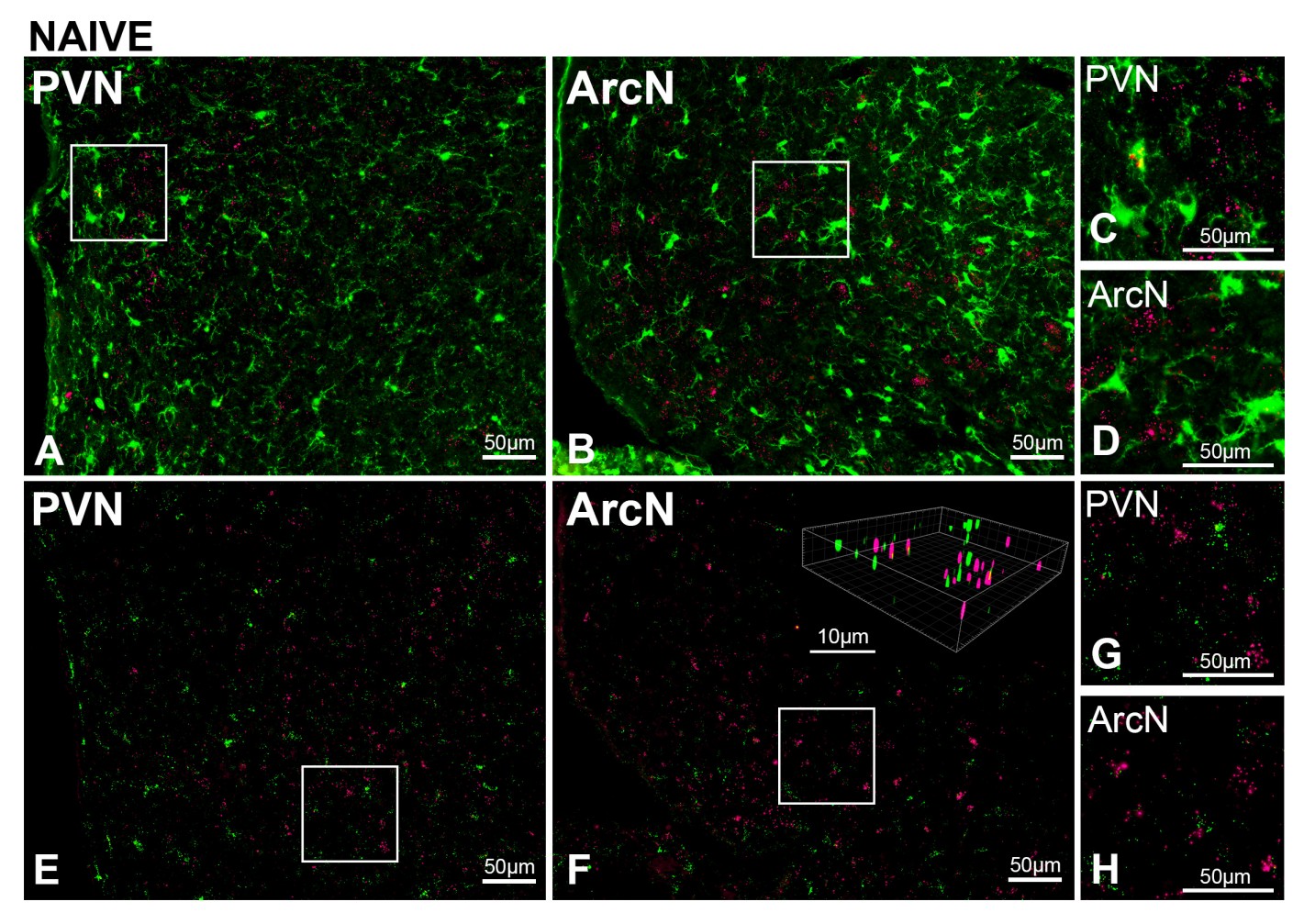

**Figure 10.** PVN LepR are rarely expressed in PVN microglia in untreated (naïve) male rats. Top. Representative images (from n = 2) showing Iba-1 (green, ihc) and LepR (red puncta, FISH) in PVN (**A**) and ArcN (**B**); higher magnifications of boxed subsections show rare overlap of iba-1 and LepR in the PVN (**C**), but not ArcN (**D**). Bottom. Representative images (from n = 4) Iba-1 (green, FISH) and LepR (red puncta, FISH) in PVN (**E**) and ArcN (**F**); higher magnifications showing rare overlap of iba-1 and LepR in the PVN (**G**) and ArcN (**H**). Insert: 3D image in **F** constructed using Imaris.

astroglial glutamate uptake. Indeed, icv leptin decreases glutamate uptake transporter expression, and leptin inhibits glutamate uptake in astroglial cultures (*Fuente-Martín et al., 2012*). Alternatively, activated microglia (secondary to the injection) could release cytokines to inhibit astroglial glutamate uptake (*Tilleux and Hermans, 2007*). In either case, increased local glutamate concentrations would stimulate PVN presympathetic neurons (*Stern et al., 2016*; *Bardgett et al., 2014b*).

We tested whether astroglia express LepR, in both untreated rats (PVN and ArcN) and PVN leptin/aCSF injected (n = 3). Colocalization of the LepR with GFAP-ir (to identify astroglial cells) was never observed (*Figure 12*).

Our next approach was pharmacological. Dihydrokainic acid (DHK) is a selective inhibitor of the most prevalent glutamate uptake transporter in brain, GLT-1 (EAAT2) (*Tilleux and Hermans, 2007*). Our rationale was that if leptin inhibited glutamate uptake by GLT-1, then the sympathoexcitatory action of subsequent DHK injection would be lessened. As shown in *Figure 13A*, bilateral PVN DHK injections elicited a modest increase in LSNA (55 ± 9%) and MAP in otherwise untreated rats. Interestingly, 90 min after aCSF injections, the responses to DHK were smaller (*Figure 13B*; p<0.05), suggesting that the injections themselves were sufficient to inhibit astroglial glutamate uptake, likely by local microglial activation. PVN leptin increased LSNA after 90 min, at which time DHK produced a similarly small increase in LSNA (28 ± 8%), as after aCSF injections (26 ± 4%) (*Figure 13C*). Finally,

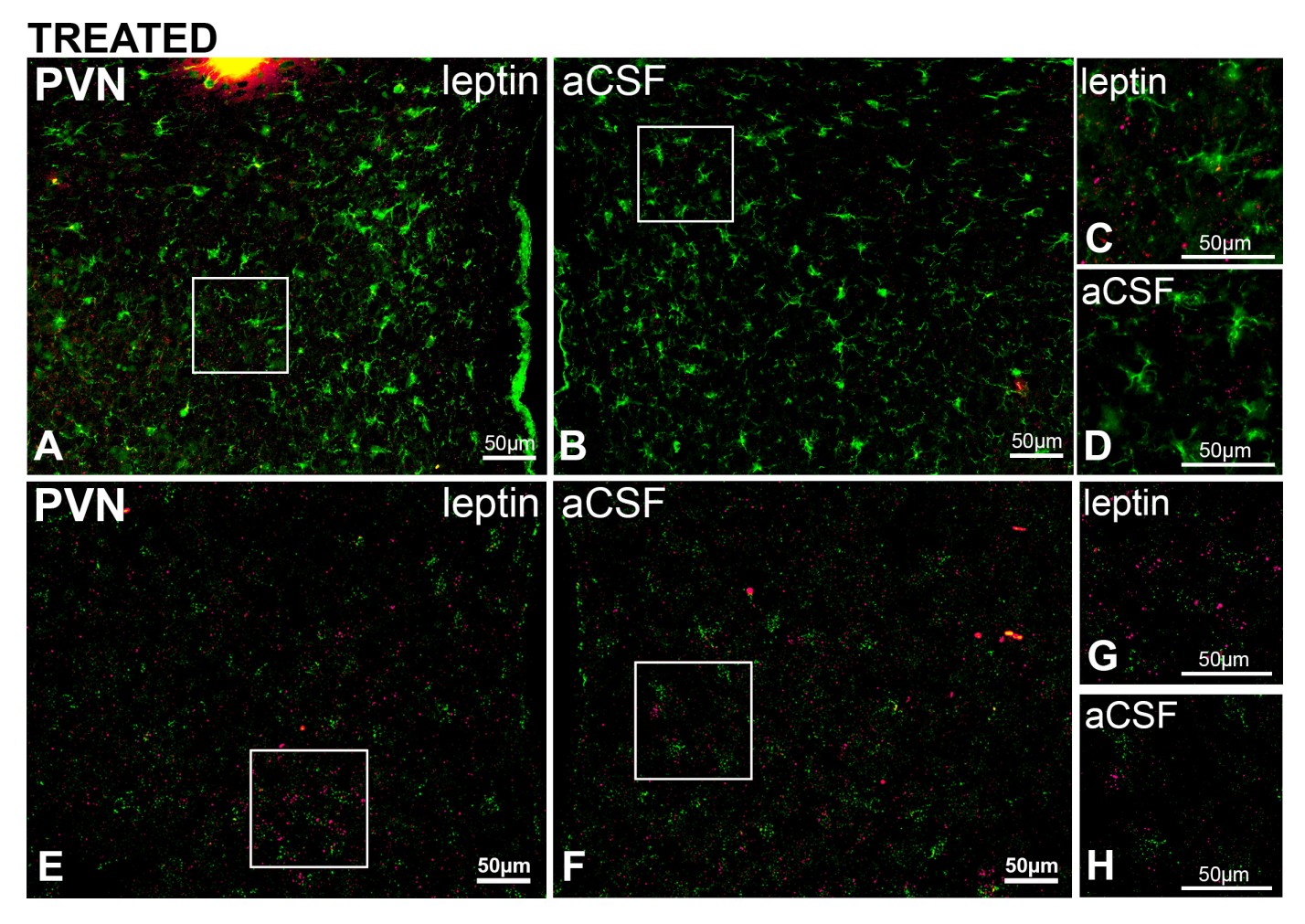

**Figure 11.** PVN LepR are rarely expressed in PVN microglia in rats treated with Leptin/aCSF. Top. Representative images (from n = 3) showing PVN Iba-1 (green, ihc) and LepR (red puncta, FISH) in leptin-treated (**A**) and aCSF-treated (**B**) rats; right images are magnifications showing rare overlap of iba-1 and LepR from the leptin-treated (**C**) but not the aCSF-treated (**D**) sides. Bottom. Representative images (from n = 2) showing PVN Iba-1 (green, FISH) and LepR (red puncta, FISH) in leptin-treated (**E**) and aCSF-treated (**F**); **G** and **H** are higher magnifications showing limited overlap of iba-1 and LepR in the PVN.

aCSF injections 90 min after PVN leptin were not associated with further significant increases in LSNA (the response had stabilized as in *Figure 1*; *Figure 13D*). Collectively, these results indicate that the reduced DHK response after PVN leptin was due to the injection itself (same as after aCSF); therefore, these data do not support the hypothesis that leptin directly or indirectly inhibits astroglial GLT-1 and glutamate uptake, thereby activating PVN pre-sympathetic neurons.

## PVN LepR are expressed in glutamatergic neurons

Our next target was PVN neurons. We used VGlut-2 FISH to identify PVN neurons, because most neurons in PVN are glutamatergic [GABAergic neurons are rare (*Pyner, 2009*)], and because the vast majority of pre-sympathetic neurons in the PVN are glutamatergic (*Stocker et al., 2006*). We also specifically identified pre-sympathetic neurons, by injecting the retrograde tracer cholera toxin B (CTB) into the RVLM. As shown in *Figure 14*, LepR were commonly found in glutamatergic neurons. However, only a small fraction of neurons positive for CTB also expressed the LepR (10–20%) (*Figure 14C,E*).

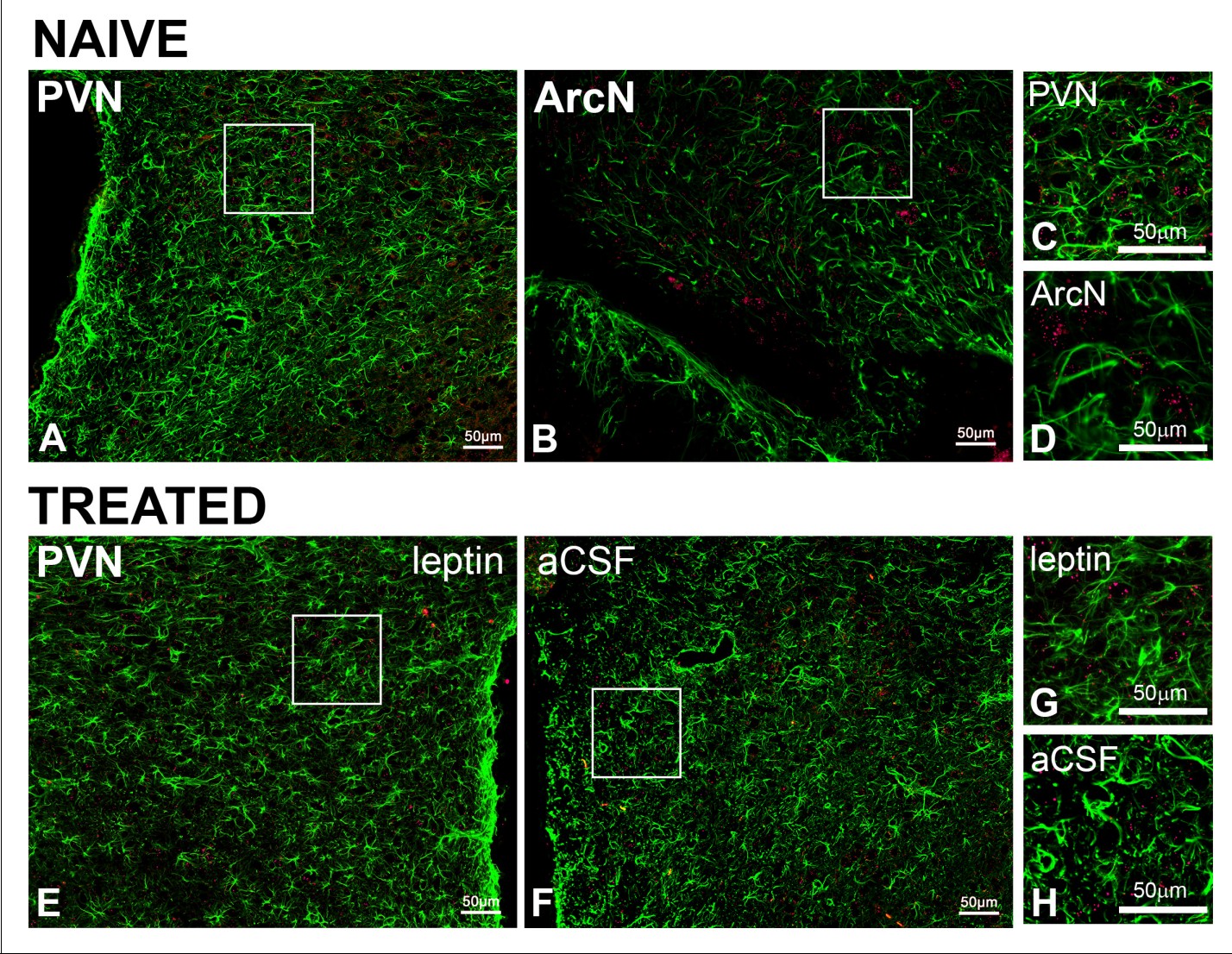

**Figure 12.** PVN LepR are not expressed in astroglia in naïve or PVN leptin/aCSF treated rats. Representative images showing GFAP (green, ihc) and LepR (red, FISH) from naïve (**A-D**, n = 6) or treated (**E-H**, n = 4) rats. In untreated rats, images from the PVN (**A and C**) and the ArcN (**B and D**), are shown. Representative images from the leptin-treated (**E and G**) and aCSF-treated (**F and H**) sides of the PVN are shown. **C, D, G**, and **H** are higher magnification views of the boxed areas in **A, B, E**, and **F**.

## PVN glutamatergic neurons are excited by leptin

To establish a functional role for LepR in PVN neurons, we selectively expressed the calcium-dependent fluorescent protein GCaMP6s in glutamatergic neurons, by injecting a cre-dependent AAV containing GCaMP6 into the PVN of VGlut-2 cre mice (*Figure 15*). Hypothalamic coronal slices containing the PVN were first treated with the GABA$_A$ antagonist, bicuculline (BICC), to eliminate the possibility that leptin also activates inhibitory GABAergic neurons that surround and project to the PVN (LepR are abundantly expressed just dorsal to the PVN, unpublished observations); as a control, some slices received BICC alone. Fifteen min later, PVN slices were incubated with leptin (or aCSF) for 90 min, and then treated with a pulse of NMDA, to assess neuronal viability.

As previously reported (*Schafer et al., 2018*), the calcium fluorescent signal tended to slowly decrease in slices treated with BICC alone (*Figure 15B*). In these slices, the calcium signal exhibited regular and entrained bursts, every few min (*Figure 15B*). Leptin significantly increased the number of activated neurons (*Figure 15A*), either when comparing the total population of BICC+aCSF versus BICC+leptin glutamatergic neurons or when comparing the % activated in each slice (*Figure 15C*).

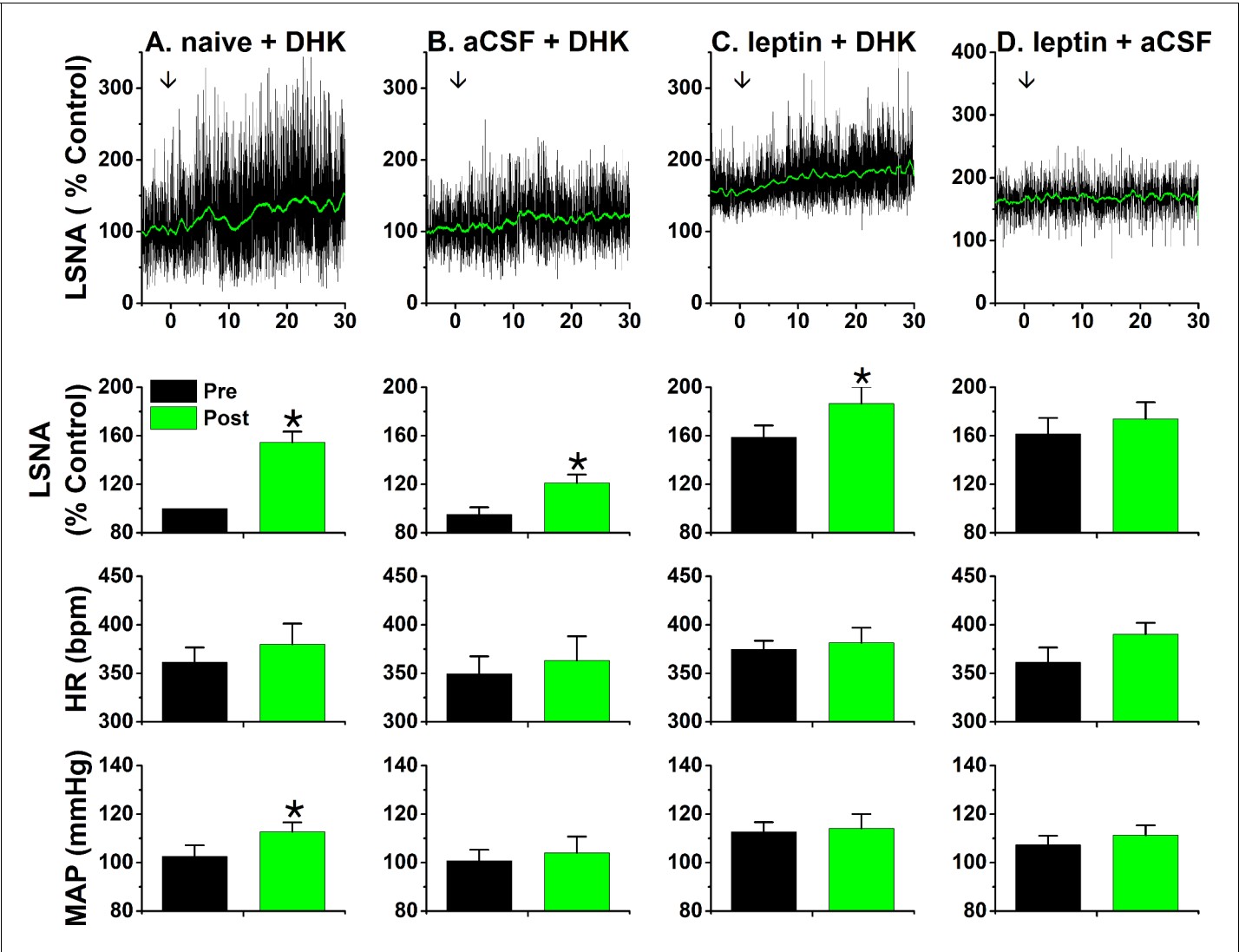

**Figure 13.** PVN leptin does not reduce the sympathoexcitatory response to PVN DHK. Top, representative experiments; Bottom, grouped data. (A) In untreated rats, bilateral PVN DHK increased LSNA (55 ± 9%) and MAP (p<0.05; n = 6). (B) Two hours after bilateral nanoninjections of aCSF, the response to DHK (26 ± 4%) was reduced (p<0.05; n = 5). (C) After two hours, leptin significantly increased LSNA (p<0.05), but subsequent PVN DHK increased LSNA (28 ± 8%; n = 6) similarly as after PVN aCSF. (D) LSNA was increased two hours after PVN leptin, and did not increase further after PVN nanoinjections of aCSF (n = 5). *: p<0.05, compared to baseline (Pre); †: p<0.05 leptin compared to icv aCSF at the same time; ‡: p<0.05, DHK after aCSF compared to DHK after untreated (naïve).

Moreover, in parallel to the sympathoexcitatory effect of PVN leptin, the increase in calcium signal in leptin-treated slices was slowly developing (*Figure 15A*). Interestingly, slices treated with leptin never exhibited the pulsatile calcium signal observed in slices treated with BICC alone (*Figure 15A*).

## PVN LepR are expressed in TRH neurons

Previous studies indirectly suggested that TRH neurons express LepR, since exogenous icv leptin administration increased p-STAT expression in these cells (*Perello et al., 2006*; *Harris et al., 2001*). *Figure 16* provides the first direct evidence for the coexistence of LepR in TRH neurons in otherwise untreated animals. Importantly, some LepR-expressing TRH neurons were also positive for CTB (*Figure 17*), retrogradely transported from the RVLM. These anatomical data suggest that leptin excites TRH presympathetic neurons that project to the RVLM. To test this hypothesis, we next determined if injection of TRH into the RVLM increases LSNA. We also determined if injection of TRH into the DMH, which houses presympathetic neurons that project to both the RVLM and the Raphe Pallidus

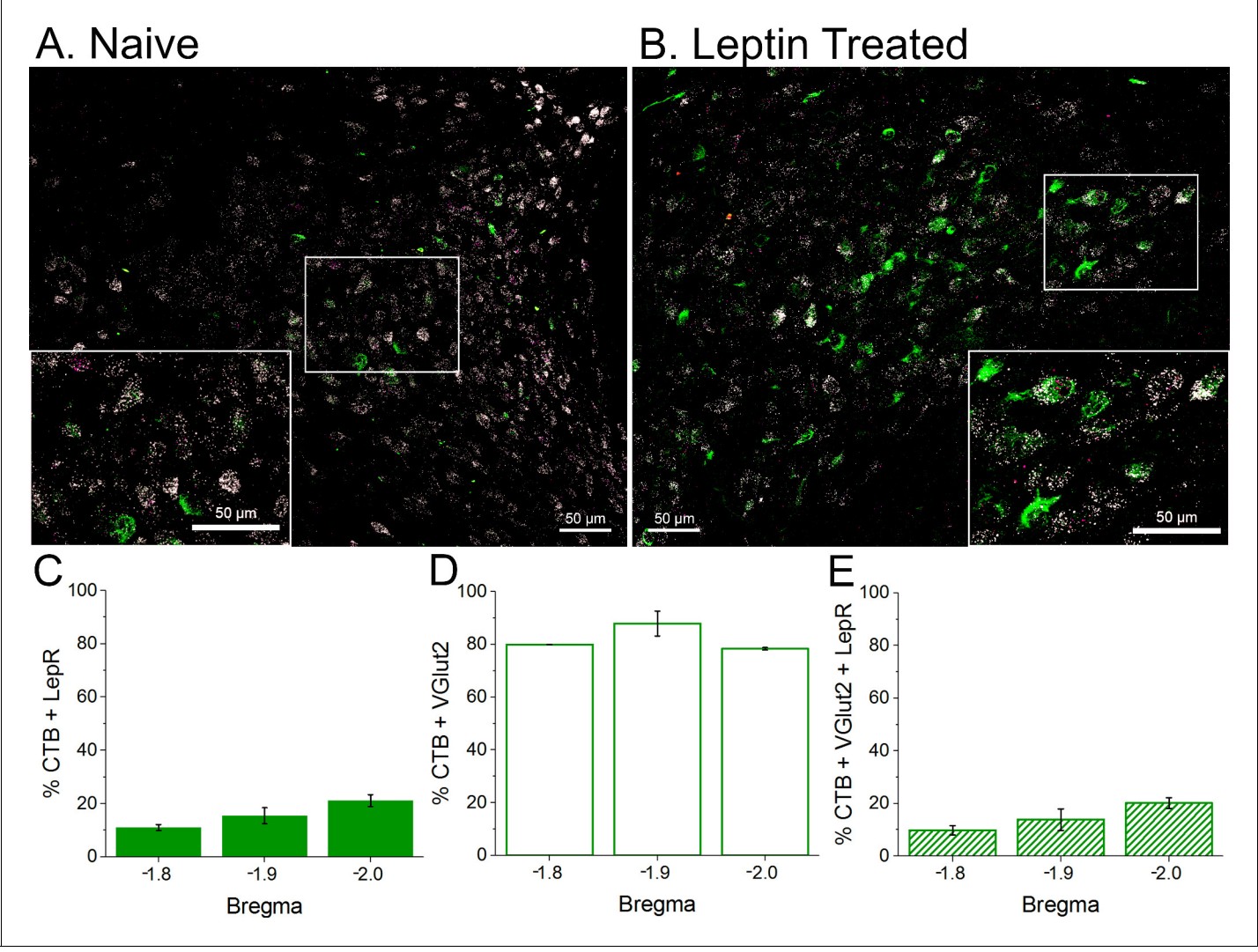

**Figure 14.** LepR are expressed in glutamatergic neurons, some of which project to the rostral ventrolateral medulla (RVLM; contain CTB).
Top. Representative sections from an untreated rat (**A**) and leptin-treated rat (**B**) showing that LepR (Red punta, FISH) are expressed in glutamateric neurons (white; FISH), some of which contain the retrograde tracer CTB (green; project to the RVLM). Bottom. Three animals were quantified (2 leptin/ aCSF treated and one untreated). (**C**) A small fraction (10–20%) of CTB containing neurons express the LepR. (**D**) Most CTB containing neurons are glutamatergic. (**E**) A similar fraction of CTB neurons that express the LepR also express VGlut2. Therefore, essentially all neurons that contain CTB and express LepR are glutamatergic.

(RPa), increases LSNA and BAT SNA. RVLM TRH, but not aCSF, increased LSNA, HR, and MAP (*Figure 18A*). In parallel, DMH TRH increased LSNA, HR, and MAP, as well as BAT SNA, BAT temperature, and $CO_2$; again, DMH aCSF was without effect on these variables (*Figure 18B,C*).

## Discussion

The PVN is a major autonomic integrative hub. While leptin is a well-established sympathoexcitatory hormone, whether or how PVN leptin increases SNA was unknown, since PVN LepR are undetectable or sparse. We show that PVN leptin slowly, but dose-dependently, increased LSNA; PVN leptin also activated BAT SNA. Leptin became effective, in part because it induced the expression of its own receptor. PVN LepR were not expressed in astroglia, and only rarely in microglia; instead, LepR were found in glutamatergic neurons, some of which project to the RVLM. Using GCaMP6, we further show that PVN leptin excited glutamatergic neurons, which contributed to its local excitatory effects on LSNA. The expression of PVN LepR largely in TRH neurons, the importance of PVN leptin in

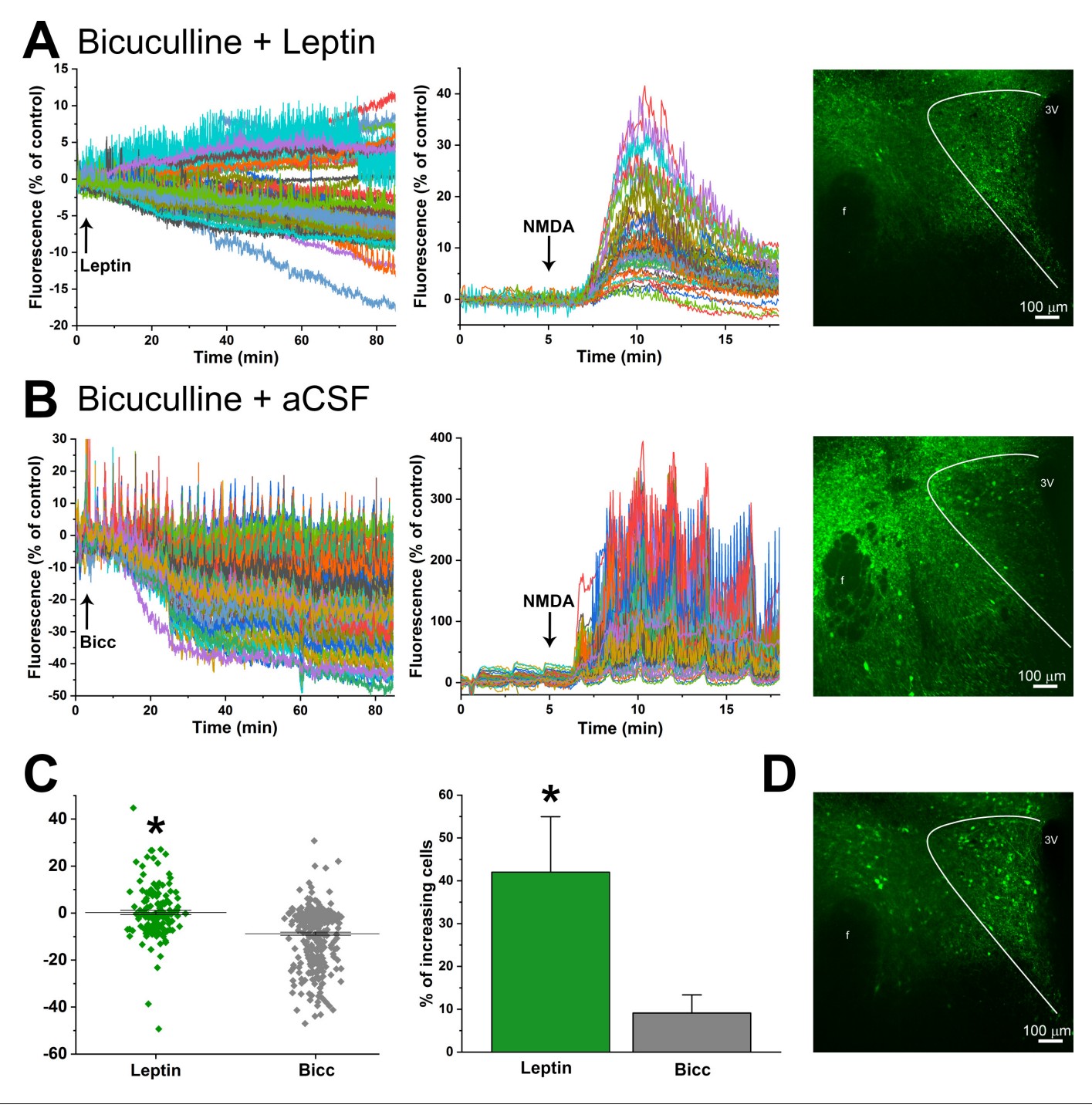

**Figure 15.** Leptin activates PVN glutamatergic neurons. (**A**) Left. Representative experimental tracing showing that PVN glutamatergic neuronal activity (GCaMP-6 fluorescence) both increases and decreases following treatment of a PVN slice with leptin+bicuculline (after 15 min pretreatment with bicuculline). Middle. Same experiment showing response to NMDA. Right. Maximum projection of fluorescence after 90 min treatment with leptin. (**B**) Representative experimental tracing showing that PVN glutamatergic neuronal activity (GCaMP-6 fluorescence) largely decreases following treatment of a PVN slice with aCSF+bicuculline (after 15 min pretreatment with bicuculline). Middle. Same section showing response to NMDA. Right. Maximum projection of fluorescence after 105 min treatment with bicuculline. (**C**) Summary of neuronal responses from leptin+bicuculline (Leptin, n = 4) and aCSF+bicuculline (Bic; n = 3) treated slices. Left summarizes all neurons from all experiments. *: p<0.0001, unpaired t-test. Right shows the percentage of neurons exhibiting increased GCaMP-6 fluorescence from all experiments (n = 4 leptin; n = 3 Bic). *: p<0.05, unpaired one-tailed t-test. (**D**) Maximum projection of the slice in **A** after treatment with NMDA.

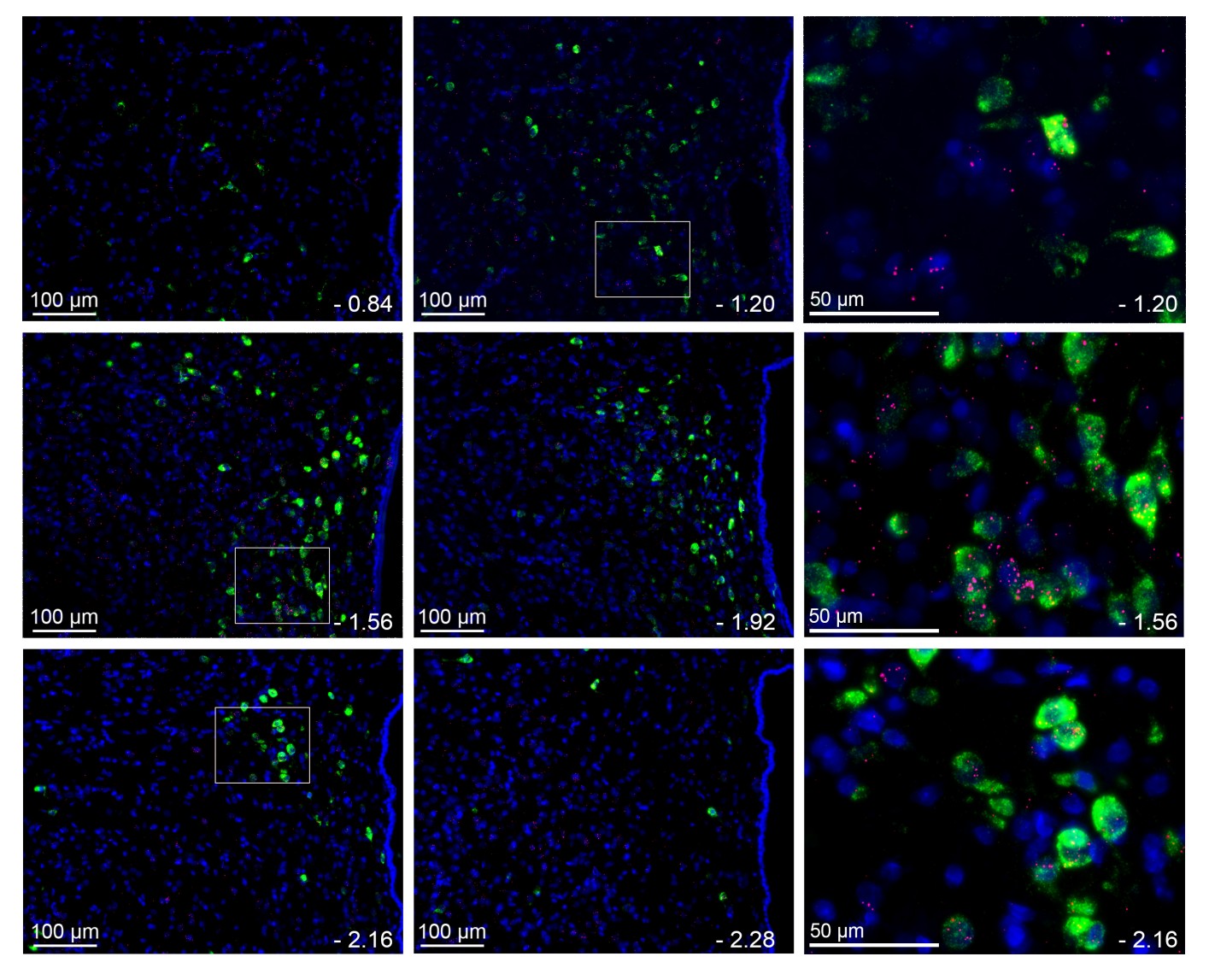

**Figure 16.** LepR are commonly, but not exclusively, found in TRH neurons. Representative images (n = 3) from several levels of the PVN (−0.84 to −2.28 from bregma) showing the expression of LepR (red puncta) in TRH neurons (green). Blue is DAPI nuclear stain. Right panel shows higher magnification from selections indicated by white boxes to the left.

supporting and setting the HPT axis (*Hollenberg, 2008*; *Joseph-Bravo et al., 2015*; *Nillni, 2010*), and the sympathoexcitatory actions of TRH in the DMH and RVLM provide homeostatic context for the actions of PVN leptin on energy balance, in particular, on energy expenditure, and on blood pressure regulation.

## PVN leptin increases SNA

It is well established that exogenous leptin enhances TRH secretion and upregulates the HPT axis via an action in the PVN; conversely, decreases in endogenous leptin during fasting suppresses PVN TRH production [for reviews, see *Hollenberg, 2008*; *Joseph-Bravo et al., 2015*; *Nillni, 2010*]. Previous studies also hinted that PVN leptin might increase SNA, since microinjection of leptin into the PVN increased arterial pressure (*Marsh et al., 2003*; *Montanaro et al., 2005*). Our ability to detect a dose-dependent sympathoexcitatory action of PVN leptin relied on a prolonged observation period, since the response was slowly developing, likely in part due to leptin's action to increase the expression of its own receptor. PVN leptin was also found to enhance baroreflex control of LSNA

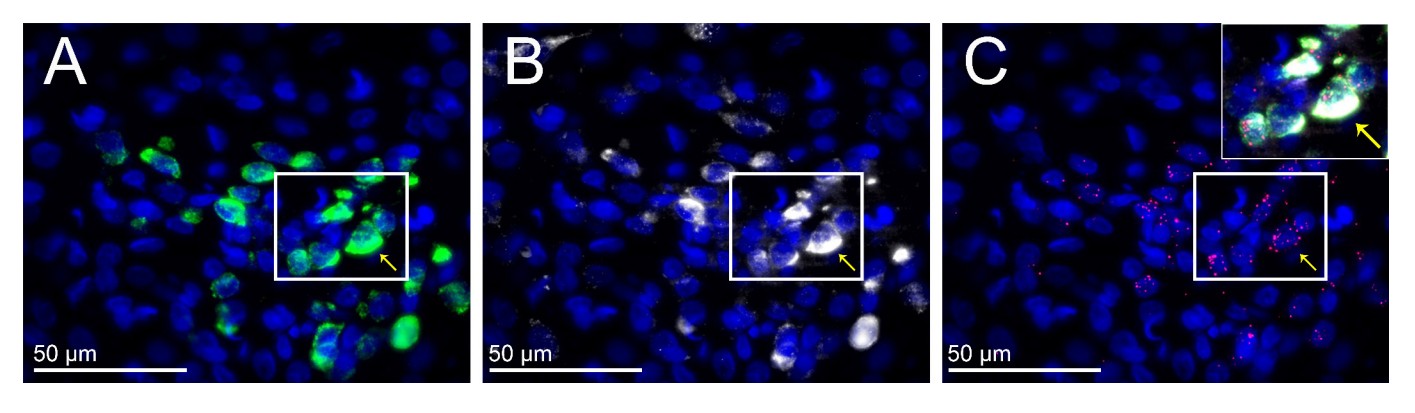

**Figure 17.** LepR are expressed in TRH neurons that project to the RVLM. Representative image from one rat that received RVLM CTB showing that many CTB-containing PVN neurons (white) express TRH (green) and the LepR (red puncta), although CTB-containing but TRH negative, and TRH-containing but CTB negative, neurons were also observed. (**A**) TRH expression; (**B**) CTB-labeled neurons; (**C**) LepR. Insert is composite of the boxed areas in **A**, **B**, and **C**. Arrows point to a TRH neuron that projects to the RVLM (contains CTB) and expresses the LepR.

and HR, similarly to icv leptin (except the LSNA baroreflex minimum was not enhanced), suggesting that part of the effects of icv leptin is via a direct action in the PVN. In addition to LSNA, PVN leptin also activated BAT SNA, and in cooled animals this response was heightened sufficiently to significantly increase BAT temperature. Collectively, these actions are physiologically relevant. By increasing LSNA and its baroreflex regulation, meal-induced increments in leptin, via sympathetic activation, enhance glucose uptake by skeletal muscle (*Seoane-Collazo et al., 2015*; *Shiuchi et al., 2017*). By increasing HR, leptin also contributes to the postprandial increases in cardiac output (*Dagenais et al., 1966*) and the delivery of glucose to muscle vascular beds. Its action to increase BAT SNA reinforces the effects of PVN leptin to upregulate the HPT axis and increase plasma thyroxine levels, which can directly activate BAT.

In females, icv leptin increases LSNA only during proestrus (*Shi and Brooks, 2015*). Estrogen also amplifies the anorexic effect of leptin (*Clegg et al., 2006*), via ERα (*Frank et al., 2014*), although the mechanisms of this synergism remain unclear (*Kim et al., 2016*). The present results that nano-ninjections of leptin into the PVN increased LSNA only in the presence of proestrus levels of estrogen suggest that the interaction between leptin and estrogen is locally within the PVN. However, the estrogen receptors involved [the PVN expresses largely ERβ (*Oyola et al., 2017*; *Laflamme et al., 1998*)] and the nature of the interaction require further study.

## Leptin increases the expression of its own receptor in PVN

Previous studies utilizing ISH or a LepR reporter mouse described PVN LepR expression as weak (*Guan et al., 1997*; *Elmquist et al., 1998*; *Scott et al., 2009*). Our data confirm the weak signal revealed by classical ISH. However, alternate approaches, such as quantification of leptin-induced p-STAT3 induction (*Huo et al., 2004*; *Perello and Raingo, 2013*) or electrophysiological responses to leptin (*Ghamari-Langroudi et al., 2010*; *Powis et al., 1998*) suggest that PVN LepR are present in physiologically relevant levels. The present FISH findings confirm that LepR expression is scarce, compared to other sites like the ArcN. Moreover, a detailed analysis of the PVN revealed that LepR are present throughout the parvocellular PVN, but are particularly concentrated in the more caudal regions, in both rats and mice.

We hypothesized that significant responses to PVN leptin can be observed, despite few LepR, because leptin induces the expression of its own receptor, based on the following previous observations: 1) Leptin increases LepR expression in cultured microglia (*Tang et al., 2007*) and in the ArcN [cell type unknown (*Mitchell et al., 2009*)]; 2) icv leptin slowly increases SNA and blood pressure [*Figures 1*, *4*, *5* and *8* and (*Shi et al., 2015*; *Li et al., 2013*; *Shi and Brooks, 2015*)]; and 3) prior icv leptin administration amplifies the hypertension elicited by a second leptin challenge (*Han et al., 2016*). Our results support this hypothesis and further demonstrate that the LepR induction is restricted to the more caudal levels of PVN, even though its presence can be detected in more

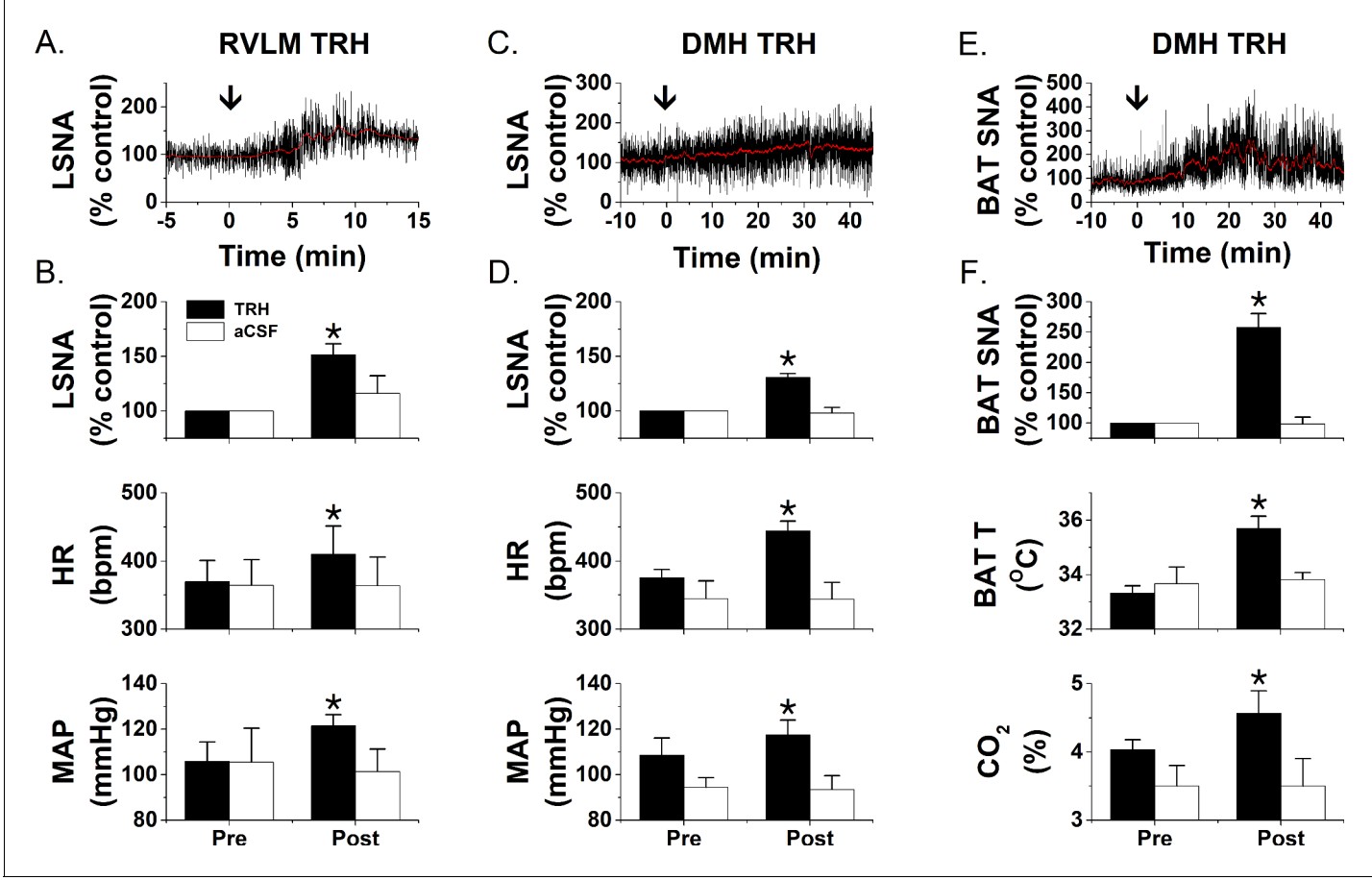

**Figure 18.** TRH increases LSNA and BAT SNA. Nanoinjections of TRH into the RVLM (**A and B**; n = 5) increase LSNA, HR, and MAP, whereas RVLM injections of aCSF (**B**; n = 3) have no effects. Nanoinjections of TRH into the DMH increase LSNA, HR, and MAP (**C and D**; n = 6) as well as BATSNA, BAT temperature, and expired $CO_2$ (**E and F**; n = 5). Again, DMH aCSF injections (**D and F**; LSNA, n = 6; BATSNA, n = 4) were without effect. *: p<0.05 compared to basal.

rostral regions. This finding is significant, because it helps explain why relatively small but sustained increases in endogenous leptin can inhibit food intake (for example), but given acutely much higher exogenous concentrations/doses are required. It will be important to test if leptin exerts the same effect in neurons in other brain sites enriched with the LepR. Candidate sites in the brainstem include the RVLM and NTS, in which acute leptin administration can elicit sympathoexcitation, but only with very high doses (*Mark et al., 2009*; *Barnes and McDougal, 2014*).

## PVN leptin-induced sympathoexcitation involves glutamate

We next sought to delineate the cellular-molecular mechanisms by which leptin in PVN increases SNA. Previous studies demonstrated that prior icv administration of KYN, to broadly block hypothalamic ionotropic glutamate receptors, prevented the subsequent pressor response to icv leptin (*Yu and Cai, 2017*) and that PVN KYN and PVN SHU9119 (inhibits local MC3/4R) each partially reversed the sympathoexcitatory responses to icv leptin (*Shi et al., 2015*). The present results show that KYN, but not SHU9119, partially reversed the local sympathoexcitatory effects of PVN leptin, and therefore suggest a local interaction between leptin and glutamate, but not α-MSH, in its sympathoexcitatory actions. However, the cell types involved remained unknown. We therefore systematically investigated the role of PVN LepR in microglia, astroglia, and glutamatergic neurons in the sympathoexcitatory effects of PVN leptin, since one or more of these populations could contribute.

We considered microglia a prime candidate, based on several reports identifying LepR in microglia both in vitro and in vivo, as well as the work of Rothwell and colleagues (*Luheshi et al., 1999*; *Pinteaux et al., 2007*) showing that leptin inhibits food intake via direct microglial activation and

release of IL-1β. However, while *lepr* mRNA was detectable in ArcN microglia, using RNAScope to amplify signals, they were rarely seen in PVN microglia. Similarly, using both anatomical and pharmacological approaches, we were unable to identify LepR in astroglia. A similar paucity of LepR in non-neuronal cells was recently reported in mice (*Yuan et al., 2018*). Therefore, we conclude that glia likely do not play a major role in the sympathoexcitatory responses to PVN leptin. Explanations for our inability to detect LepR in PVN glia, despite reports to the contrary in other brain regions, include: 1) Several studies used ihc to survey LepR in astroglia, even though antibodies against the LepR are notoriously nonspecific. Indeed, we found extensive co-localization of LepR-ir and GFAP-ir in the PVN with a commonly employed LepR antibody (Madden and Brooks, unpublished observation), but failed to confirm this co-localization using FISH. 2) Microglial function and phenotype clearly depend on the local environment, which varies throughout the brain. 3) More sustained increases in leptin may directly or indirectly activate microglia, such as with a HFD (*Gao et al., 2014*). 4) Given the occasional occurrence of microglial LepR that we observed in the PVN, it remains possible that under some circumstances, such as with sustained inflammation, activated PVN microglia may induce expression of LepR.

Instead of glia, LepR were expressed in PVN glutamatergic neurons (*Figure 14*). We used select expression of GCaMP6 to test the functional significance of this expression, since the calcium signal correlates well to membrane potential and action potential generation (*Blanco-Centurion et al., 2019*). We found that a significant fraction (40%) of PVN glutamatergic neurons exhibited a slowly developing increase in fluorescence in the presence of leptin. The vast majority of PVN-to-RVLM neurons are glutamatergic (*Stocker et al., 2006*), and we show that a subset of these neurons express the LepR. Therefore, PVN leptin may increase SNA by directly exciting PVN presympathetic neurons that project to RVLM (*Figures 14* and *17*). In this scenario, leptin may also enhance the excitation imparted by glutamatergic inputs, as shown previously in the hippocampus (*Moult and Harvey, 2009*). Alternatively, the PVN contains many glutamatergic interneurons (*Csáki et al., 2000*), which mediate the local effects of norepinephrine (*Boudaba et al., 1997*) and corticotropin releasing hormone (*Jiang et al., 2018*) within the PVN. Thus, in addition to a direct action, PVN leptin may activate presympathetic neurons indirectly via local glutamatergic interneurons (*Figure 19A*).

An unexpected finding was that BICC treatment alone induced uniform pulsatility in the GCaMP6 calcium signal. A key feature was that the neuronal calcium pulsation was largely entrained among

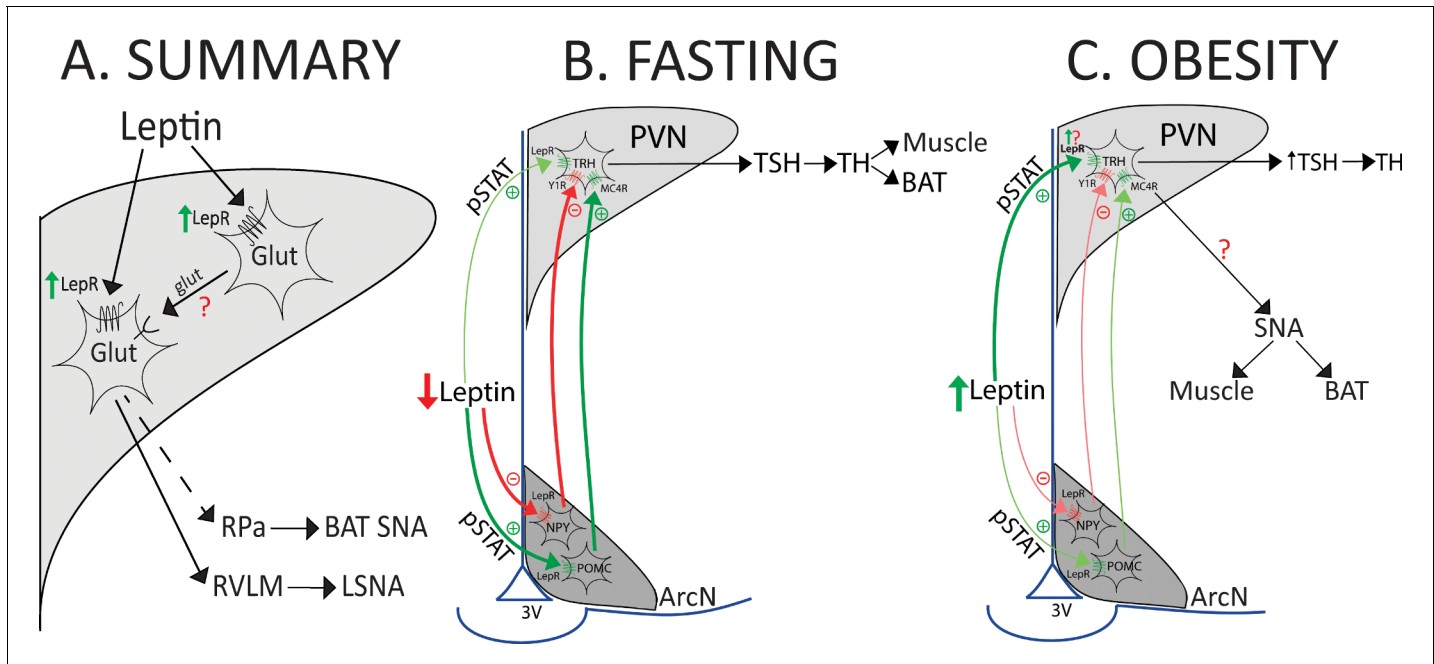

**Figure 19.** Models that summarize the results and hypothesize potential roles during fasting or obesity. (**A**) Summary of findings. (**B and C**) Models illustrating hypothesized role of PVN LepR in the regulation of the HPT axis and LSNA and BATSNA with fasting and obesity.      See text for details.

neurons. This finding is reminiscent of the effect of PVN nanoinjections of BICC to induce relatively slow, synchronized bursting of several sympathetic nerves (*Kenney et al., 2003*; *Kenney et al., 2001*). The SNA bursting pattern was eliminated by pretreatment with KYN (*Kenney et al., 2003*), indicating that local glutamatergic mechanisms are involved. Therefore, our finding that the bursting was not observed in the presence of leptin, reinforces the conclusion that leptin engages local glutamatergic mechanisms in its activation of PVN presympathetic neurons.

## LepR are expressed in TRH neurons

The HPT axis is suppressed during fasting to decrease the basal metabolic rate and energy expenditure (for reviews, see *Hollenberg, 2008*; *Joseph-Bravo et al., 2015*; *Nillni, 2010*; *Figure 19B*). Leptin replacement prevents the fasting-induced decreases in TRH and thyroid hormones, indicating that the fall in leptin mediates the HPT axis suppression. Administration of leptin to fasted animals evokes p-STAT expression in both the PVN and the ArcN. In the PVN, leptin induces p-STAT mainly in TRH neurons (*Perello et al., 2006*), suggesting that leptin can directly excite these neurons. The present results demonstrating the presence of LepR in TRH neurons establish that leptin is capable of direct activation. In addition to this direct effect, leptin also alters the activity of two major ArcN neuronal types that project to the PVN: it activates POMC neurons, which release α-MSH in the PVN, and inhibits Neuropeptide Y (NPY) neurons (*Figure 19B,C*). Because α-MSH excites, and NPY inhibits, TRH neurons, leptin can also stimulate TRH neurons indirectly, via the ArcN. Previous studies indicate that the indirect pathway mediates the leptin-mediated falls in TRH during fasting, rather than the direct pathway (*Hollenberg, 2008*; *Nillni, 2010*; *Figure 19B*). The present results documenting far fewer LepR in PVN TRH neurons compared to the ArcN provide a mechanistic explanation for the dominance of the ArcN in this homeostatic suppression. Nevertheless, elimination of LepR from PVN neurons does decrease basal body and BAT temperatures (*Cakir et al., 2019*). Because the sympathetic nerve innervating BAT is generally silent in animals and humans at neutral temperatures, the decrease in BAT temperature is not likely mediated by the sympathetic nervous system, despite our finding that PVN leptin activates BAT SNA. Instead, it appears that PVN LepR are required for optimal function of the HPT axis and production of thyroid hormones, which directly sensitize BAT to other excitatory inputs, like SNA.

PVN TRH neurons project to the DMH (*Wittmann et al., 2009*), which richly expresses TRH receptors (*Manaker et al., 1985*; *Calzá et al., 1992*; *Heuer et al., 2000*) and sends major projections to the RPa, which houses BAT sympathetic premotor nerves. The fact that TRH can increase brown adipose thermogenesis (*Griffiths et al., 1988*), in particular via an action in the DMH (*Shintani et al., 2005*), has been known for decades. Moreover, TRH increases blood pressure (*García and Pirola, 2005*), and some PVN inputs to the RVLM, a major presympathetic hub, contain TRH (*Lee et al., 2013*). We further show that TRH neurons that project to the RVLM express LepR. However, whether and where TRH increases BAT or lumbar SNA had not been previously investigated. Here we show that DMH and RVLM TRH each elicit increases in LSNA, and DMH TRH increases BAT SNA, in association with an increase in BAT temperature and $CO_2$ production. These findings not only reinforce the known effects of TRH on energy expenditure and blood pressure regulation, but also provide a substrate for further studies of the neuronal circuitry by which TRH, via an interaction with leptin, increases energy expenditure and vasomotor SNA.

## Perspectives

Our conclusions evolved from data obtained from both rats and mice, albeit largely from rats. Rats and mice exhibit subtle differences in the spatial interrelationships among PVN subnuclei (*Biag et al., 2012*; *Simmons and Swanson, 2009*) and in brain LepR expression (*Scott et al., 2009*). Therefore, whether our findings in rats can be extrapolated to mice, or more importantly to humans, remains to be determined. Nevertheless, several pertinent features are similar between the species, including: 1) low PVN LepR expression throughout the PVN compared to the ArcN, although more highly expressed in medial to caudal PVN [*Figures 6* and *7*]; 2) TRH expression throughout the PVN, with neuroendocrine neurons concentrated in caudal PVN and non-neuroendocrine cells more commonly identified in rostral PVN (*Simmons and Swanson, 2009*; *Fitzsimmons et al., 1994*); 3) preponderance of preautonomic neurons in medial and caudal PVN (*Biag et al., 2012*; *Stocker et al., 2004*); 4) slow responses to PVN leptin [SNA responses in rats (*Figure 1*) and glutamatergic neuronal

calcium responses in mice (*Figure 15*)]; 5) paucity of LepR in non-neuronal cells [*Figures 10–13* and (*Yuan et al., 2018*)]; and 6) involvement of glutamatergic neurons in the effects of PVN leptin (*Figures 9* and *15*). However, key findings require replication in mice (and humans), in particular the ability of leptin to induce the expression of its own receptor in vivo.

We suggest that the slowly developing nature of the increases in lumbar and BAT SNA induced by PVN leptin are due in part to its upregulation of LepR. However, because the increases in SNA arise both before and more than the upregulation of LepR, it appears that other factors contribute. We speculate that these downstream events may include cellular synergism between PVN glutamate and leptin signaling (*Figure 9*), the delayed synthesis and release of TRH in downstream brainstem targets and synergism between TRH and fast-acting neurotransmitters like glutamate, and a complex interplay between circuits engaged by PVN leptin. These enhancing effects may be amplified even more when leptin increases systemically and acts at several forebrain and brainstem sites simultaneously.

Given these new data showing that TRH increases LSNA and BAT SNA, can an increase in leptin, by upregulating its receptor, act physiologically in the PVN to increase SNA to muscle and BAT via TRH neuronal projections? Insulin is a rapid regulator of metabolism, inducing changes in glucose uptake and energy expenditure within minutes after eating. In contrast, while its metabolic partner, leptin, produces many of the same effects, the time course is slower. For example, in humans, food intake increases leptin levels, but this response takes hours to develop (*Elimam and Marcus, 2002*; *Romon et al., 1999*; *Lee and Fried, 2009*). In ob/ob mice, chronic leptin treatment only slowly (days) increases body temperature (*Fischer et al., 2016*). In parallel, unlike most other hormones like insulin, which down-regulate their receptor (*Mayer and Belsham, 2010*), we show that leptin slowly increases the expression of its receptor, at least in the PVN. Leptin treatment for one week also upregulates *lepr* mRNA in the ArcN (*Mitchell et al., 2009*), although the cell type expressing the LepR was not identified. These slow responses are ideal for the regulation of basal metabolic rate and the HPT axis. Indeed, one teleologically relevant application of this slow course is with the induction of a high fat diet or obesity (*Figure 19C*). Interestingly, rats with diet-induced obesity exhibit increased TRH and an upregulation of the HPT axis, despite leptin resistance in the main site of action of leptin, the ArcN (*Araujo et al., 2010*; *Perello et al., 2010*). Instead, the direct pathway for leptin's action on TRH neurons maintains normal or even elevated HPT activity (*Perello et al., 2010*). While the explanation for the elevated activity has not been determined, we speculate that part of the amplification involves increased TRH neuronal LepR expression (*Figure 19C*). TRH parvocellular neurons that project to the median eminence comprise a separate population from TRH neurons influenced by ArcN POMC neurons, like those that regulate SNA (*Nillni, 2010*). Therefore, in parallel, LepR in non-hypophysiotropic neurons may be up-regulated to contribute to obesity-induced sympathoexcitation (*Figure 19C*), which is dependent in part on increased leptin levels (*Hall et al., 2010*; *Bell and Rahmouni, 2016*). In support, knockdown of hypothalamic preproTRH normalizes blood pressure in hypertensive diet-induced obese rats (*Landa et al., 2007*).

## Summary

We show that PVN leptin induces the expression of its own receptor, which allows leptin to elicit a gradual increase in SNA to muscle, which will facilitate glucose uptake, and to BAT, which will increase energy expenditure. The mechanism by which PVN leptin increases LSNA involves local glutamatergic excitation, either via postsynaptic synergism of glutamatergic inputs onto PVN presympathetic neurons or by activation of glutamatergic interneurons, which will also amplify leptin's effects. Our data reveal that PVN leptin activates glutamatergic TRH neurons, which project to integrating sites important in the regulation of LSNA and BAT SNA: the DMH and RVLM. It will be important to establish whether this LepR upregulation occurs in other leptin receptive sites, as it may invoke a role for increases in leptin that are more physiologically important than previously imagined (*Flier and Maratos-Flier, 2017*). The contribution of the leptin-TRH-BAT/lumbar SNA pathway in obesity development and maintenance, in the difficulties of obese individuals in maintaining weight loss (*Rosenbaum et al., 2018*), as well as in increased SNA and hypertension, also deserve future consideration.

# Materials and methods

## Key resources table

| Reagent type (species) or resource | Designation | Source or reference | Identifiers | Additional information |
|---|---|---|---|---|
| Genetic reagent (*M. musculus*) | VGlut2-cre; C57BL/6J *Slc17a6*$^{tm2(cre)Lowl}$ | Jackson Laboratory; reference (*Vong et al., 2011*) | Stock #: 016963 RRID:IMSR_JAX:016963 | |
| Antibody | Monoclonal mouse anti-GFAP | Sigma | Catalog#: G3893 RRID:AB_477010 | Ihc (1:500) |
| Antibody | Goat anti-mouse 488 | Jackson ImmunoResearch | Catalog #: 115-545-166 RRID:AB_2338852 | Ihc (1:1500) |
| Antibody | Rabbit anti-iba1 | Wako | Catalog #: 019–19741 RRID:AB_839504 | Ihc (1:250) |
| Antibody | Goat Anti-Cholera Toxin B Subunit | List Biological Laboratories, Inc | Catalog #: 703 | Ihc (1:250) |
| Antibody | Donkey Anti-Rabbit IgG (H+L) | Jackson ImmunoResearch | Catalog #: 711-545-152 RRID:AB_2313584 | Ihc (1:1000) |
| Antibody | Donkey anti goat 488 | Jackson ImmunoResearch | Catalog #: #705-545-147 RRID:AB_2336933 | Ihc (1:1000) |
| Recombinant DNA reagent | pAAV.Syn.Flex. GCaMP6s. WPRE.SV40 | Douglas Kim and GENIE project | RRID:Addgene_100845 | |
| Commercial assay or kit | RNAScope Fluorescent Multiplex Kit | ACDBio | Catalog #: 320850 | |

## Animals

All procedures involving rats or mice were approved by the OHSU Animal Care and Use Committee. *Experimental protocols in anesthetized rats.* Male Sprague-Dawley rats (Charles River; 350–450 g) were anesthetized with isoflurane and prepared for icv infusions, PVN nanoinjections, and for measurements of MAP, HR, LSNA, and baroreflex control of LSNA and HR, as previously described (*Li et al., 2013*; *Shi et al., 2019*). In some rats, BAT temperature was monitored using a thermocouple meter (4134; Control Company, Friendswood, Texas) with a Type T needle style microprobe thermocouple (Physitemp, Clifton, NJ) inserted in the intact, left interscapular BAT fat pad. BAT SNA was recorded with a bipolar hook electrode from the central cut end of a small-diameter nerve bundle isolated from the ventral surface of the right interscapular fat pad after dividing it along the midline and reflecting it laterally. The SNA signal was then rectified and integrated in 1 s bins along with BAT temperature. After data collection, post-mortem SNA was quantified and subtracted from values of SNA recorded during the experiment. SNA was normalized to the 30 s averages before nanoinjections (% of control).

When surgery was completed, rats were transitioned over 30 min to α-chloralose anesthesia (Sigma; 50 mg/kg loading dose followed by 25 mg/kg hr continuous infusion for the duration of the experiment), while slowly withdrawing the isoflurane. In some rats, a water-perfused thermal blanket was used to maintain the skin and body temperature. Expired $CO_2$ was continuously recorded and maintained at 30–40 mmHg via artificial ventilation and by adjusting the respiratory rate and tidal volume of 100% oxygen. After 1 hr stabilization period, one of the following randomly selected protocols was performed.

## Protocol 1

### Do PVN nanoninjections of leptin increase LSNA and its baroreflex regulation?

After baseline measurements, leptin (15, 30, or 60 ng in 60 nL) or aCSF was injected bilaterally into the PVN. Measurements of MAP, HR, and LSNA or BAT SNA were collected every 30 min for 2 hr after the injections. In rats receiving the highest PVN leptin dose (60 ng) or aCSF, baroreflex curves were generated before, and 60 and 120 min after the nanoinjections. These nanoinjections were conducted in males and in ovariectomized (OVX) females implanted with an estrogen pellet that produces Proestus-levels of estrogen (*Shi and Brooks, 2015*), or a sham pellet. Uterine weight was significantly smaller (p<0.05) in OVX (0.17 ± 0.02 g) compared to OVX+E2 (0.61 ± 0.04 g) rats.

To confirm anatomical specificity of the responses to PVN leptin, similar bilateral injections were made lateral to the PVN (60 ng leptin) and also into the DMH (60 ng leptin) and ArcN (30 ng).

## Protocol 2

### Are the effects of PVN leptin reversed by blockade of PVN MC3/4R or ionotropic glutamate receptors?

Bilateral injections of leptin (30 ng in 60 nL) or aCSF were followed at 90 min by bilateral nanoinjections of the ionotropic glutamate receptor (iGluR) blocker, kynurenate (KYN; 2.7 nmol), the melanocortin type 3 and 4 (MC3/4R) receptor antagonist, SHU9119 (30 pmol), or aCSF, and LSNA, MAP and HR were continuously measured for an additional 30 min. The doses of these drugs were found previously to reverse the effects of icv leptin (*Shi et al., 2015*).

## Protocol 3

### Does PVN leptin inhibit astrocytic glutamate uptake?

Ninety min after bilateral leptin (60 ng) or aCSF injections, dihydrokainic acid [DHK; 0.12 nmol (*Stern et al., 2016*); EAAT2 (GLT-1)-selective inhibitor of L-glutamate and L-aspartate uptake] or aCSF was injected bilaterally into the PVN, and measurements of MAP, HR, and LSNA were continued for 30 min. In some rats, DHK was injected into the PVN before any other PVN injections, and measurements were again continued for 30 min.

## Protocol 4

### Does TRH increase LSNA or BAT SNA via actions in the DMH and/or RPa?

TRH (0.5 mM in 60 nL, P1319, Sigma-Aldrich St. Louis MO) or aCSF was injected bilaterally into the DMH or RVLM (3.5 mm caudal from Lamda, 1.8–2.0 mm lateral from the midline, and 8.8 mm ventral from the dura) and measurements of MAP, HR, and LSNA or BAT SNA were continued for 30 min.

*Figure 20* illustrates PVN, DMH, and RVLM injection sites.

### Baroreflex curve generation

Sigmoidal baroreflex curves were generated by first quickly lowering MAP to ~50 mmHg via iv infusion of nitroprusside (1 mg ml$^{-1}$; 20µl min$^{-1}$), followed by steadily and smoothly raising MAP to ~175 mmHg over 3–5 min by both withdrawing nitroprusside and infusing phenylephrine at increasing rates (1 mg ml$^{-1}$; 1–35 $\mu$l min$^{-1}$). Baroreflex curves relating LSNA or HR to MAP were constructed from data obtained during the increasing MAP phase from 50 to 175 mmHg. The sigmoidal baroreflex relationships were fitted and compared using the Boltzman equation: LSNA or HR = ($P_1$ – $P_2$)/[1 + exp$P_4$(MAP – $P_3$)]. $P_1$ is the maximum LSNA or HR, $P_2$ is the minimum LSNA or HR, $P_3$ is the MAP associated with the LSNA/HR value midway between the maximal and minimal values (BP50; denotes position of the curve on the x-axis), and $P_4$ is the coefficient used to calculate maximum gain, –($P_1$ – $P_2$)x$P_4$x¼, which is an index of the slope of the linear part of the sigmoidal baroreflex curve. Absolute values of gain, the maximum and minimum LSNA or HR, and the BP50 are illustrated in the figures.

### In situ hybridization (ISH)

Rats were anesthetized, instrumented for MAP, HR, and LSNA recordings as described above. Leptin (60 ng) was injected on one side of the PVN, and aCSF (to control for the surgery/injection) was

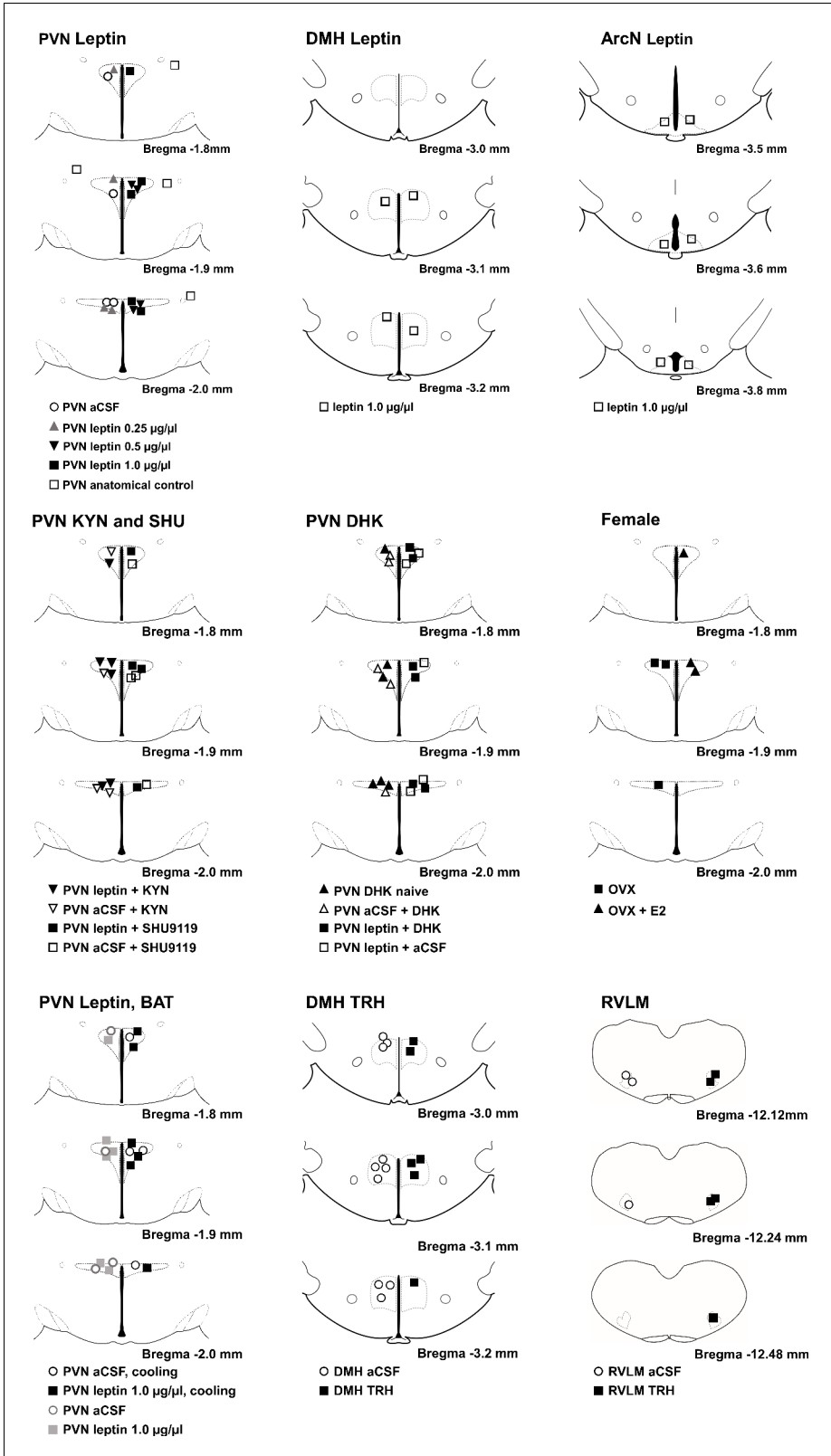

**Figure 20.** Histological maps (adapted from *Paxinos and Watson, 2007*) illustrating nanoinjection sites.

injected on the other. After 4–6 hr, the rats were euthanized, and the brains were immediately removed and snap frozen. Brains were then sectioned (20 µm) and stored at −80°C for later ISH, as previously described (*Grossberg et al., 2010*; *Scarlett et al., 2007*). Briefly, antisense $^{33}$P-labeled LepR (corresponding to bases 10–407 of rat LepR; GenBank accession number NM012596.1) and antisense digoxgenin labeled oxytocin (OT; corresponding to base pairs 67–448; GenBank accession number M25649.1) riboprobes were applied to slides, cover slipped, and incubated overnight at 55° C. The digoxigenin-labeled riboprobe for oxytocin was used to consistently select areas within the PVN for comparison between animals. Slides were then treated with RNase A, washed, and incubated with blocking buffer and antidigoxigenin fragments conjugated to alkaline phosphatase (1:250 in Tris buffer) for 3 hr at RT. After dipping in 100% ethanol and air drying, slides were dipped in NTB-2 liquid emulsion and after 11–13 days, developed and cover slipped. Signal was quantified using ImageJ. Briefly, OT-mRNA-containing cells were identified under fluorescent illumination to select five anatomically matched sections of PVN from each animal (corresponding to 1.8, 1.9, 2.0, 2.1, and 2.2 mm from Bregma) and to define the PVN region-of-interest (ROI). The silver grains were then counted in each side of the PVN, after correcting for background.

## RNAScope ISH (FISH)

Two groups of male rats were used: Untreated rats and rats two hr after PVN injections of leptin (60 ng)/aCSF, prepared as described above. Rats were deeply anaesthetized and perfused transcardially with physiological saline followed by ice cold 4% paraformaldehyde in 0.1 M sodium phosphate buffer. Brains were removed, post fixed overnight at 4°C, incubated in successive 10/20/30% sucrose gradient (each overnight), snap-frozen, sectioned (15–20 µm), mounted onto SuperFrost Plus slides (Fisher Scientific, Waltham, MA, USA), and stored at –80°C. FISH was performed in accordance with the manufacturer's instructions using the RNAScope Fluorescent Multiplex Kit (#320850; ACDBio, Newark, CA, USA) and ACDBio probes (*Table 1*) and as previously described (*Shi et al., 2019*; *Michaelis et al., 2019*). For FISH-IHC combined experiments, after FISH hybridization, slide-mounted sections were incubated overnight (4°C) with primary antibody, washed in PBS, incubated for 2 hr (room temp) with secondary antibody, washed with PBS then stained with DAPI (ACDBio kit) and coverslipped. All antibodies were diluted in 0.1 M Tris buffer and 5% normal goat serum (Sigma G9023). Sections were imaged on an AxioImager fluorescent ApoTome2 (Zeiss,Jena,Germany) with a 20 × 0.8 PlanApo objective, with exposure times set based on negative (no signal) and positive controls (optimized signal). Final images were prepared using Zen 2 (Blue Edition) software (Zeiss) and Adobe PhotoShop CS6; brightness and contrast, and in some figures, gamma, were adjusted equally for optimal visualization.

## GCaMP6

### Viral vector injections

Male C57BL/6J *Slc17a6*$^{tm2(cre)Lowl}$ mice [Jackson strain 016963 (*Vong et al., 2011*)] aged 12 weeks were used. These mice express Cre recombinase in excitatory glutamatergic neuronal cell bodies. Mice were injected with viral vectors containing Cre recombinase-activated GCaMP6s 12–14 days prior to brain extraction [pAAV.Syn.Flex.GCaMP6s.WPRE.SV40 was a gift from Douglas Kim and

**Table 1.** RNAScope probes (ACDBio).

| Target | Species | Catalog # |
| --- | --- | --- |
| LepR | Rat (Norway) | 415951-C2 |
| Vglut2 | Rat (Norway) | 317011-C2 |
| Iba1 | Rat (Norway) | 457731-C3 |
| TRH | Rat (Norway) | 406621-C1 |
| Positive Control (Polr2a (C1) and PPIB (C2), UBC (C3) | Rat (Norway) | 320891 |
| Negative Control (DapB (of *Bacillus subtilis* strain) | Rat/mouse | 320871 |
| Positive Control | Mouse | 320881 |
| LepR | Mouse | 402731-C2 |

GENIE Project (Addgene viral prep # 100845-AAV9; http://n2t.net/addgene:100845; RRID:Addgene_100845)] (*Chen et al., 2013*). The mice were anesthetized with isoflurane (2–2.5% in $O_2$), and placed in a stereotaxic instrument with a nose cone for continuous delivery of isoflurane anesthesia. A midline skin incision exposed the surface of the skull and burr holes were made. Bilateral injections (20–60 nl each) were made in the PVN with the skull flat (coordinates: 0.7–1.0 mm caudal to bregma, 0.2–0.35 mm lateral to midline, 5.0 mm ventral to dura). The nanoinjections (Toohey Pressure System IIe) were conducted over ~1 min with a glass pipette (tip diameter,~30 μm). The micropipette was left in place for 10–20 min before withdrawal. The mice were allowed to recover for at least 2 weeks before experimentation.

## Slice preparation

For each experiment brain slices were collected at the same time of day (0900 hr). Animals were rapidly decapitated, then whole brain was quickly removed and chilled in iced (0°C) sucrose-based artificial cerebrospinal fluid (aCSF, in mM/L): 208 sucrose; 2 KCl; 1 $MgCl_2$; 1.25 $NaH_2PO_4$; 10 HEPES buffer; 10 dextrose; 26 $NaHCO_3$; 2 $MgSO_4$; 1 $CaCl_2$; bubbled with 95% $O_2$, 5% $CO_2$. Hypothalamic coronal slices (200 μm thick) were cut with a vibratome (Leica, VT1000S) in chilled sucrose-aCSF. After slicing, slices were incubated in aCSF, bubbled with 95% O2, 5% CO2, for at least 1 hr before each experiment. aCSF used for incubation and imaging (in mM/L): 124 NaCl; 5 KCl; 1.44 $NaH_2PO_4$; 5 HEPES; 10 dextrose; 26 $NaHCO_3$; 2 $MgSO_4$; 2 mM $CaCl_2$; bubbled with 95% $O_2$, 5% $CO_2$ at 25°C for incubation, 32°C for imaging.

## Imaging

Slices (maintained in 32°C aCSF in a perfusion chamber) were imaged using a Yokogawa CSU-W1 on Nikon TiE spinning disk confocal microscope. All drugs were prepared in aCSF bubbled with 95% $O_2$, 5% $CO_2$. The PVN was located and injection site accuracy confirmed anatomically and by the number of neurons expressing GCaMP6 fluorescence. A series of time courses were collected, each with the following settings: 10x magnification, no binning, 488 laser power at 25%, exposure time 400 ms, and image collection every 3 s (after the initial 3 min, during which slices were imaged every 5 s). At these settings no photo-bleaching of GCaMP was detected with imaging every 3 s over two hour test exposure. Slices were perfused and drugs were added to recirculating aCSF sequentially and imaged in the following order: aCSF 15 min; aCSF-bicuculline (BICC; 2 μM/L, Tocris #2503) 15 min; aCSF-BICC-aCSF or aCSF-BICC-leptin (100 nM/L leptin) 90 min; aCSF-BICC-aCSF-N-methyl-D-aspartate (NMDA; 20 μM/L Tocris # 0114) or aCSF-BICC-leptin-NMDA 15 min; aCSF only rinse 15 min.

## Data analysis

NIS Elements v4.40 software was used. A maximum projection at 488 nm was created for the aCSF-BICC, aCSF-BICC-leptin or aCSF-BICC-aCSF, and the aCSF-BICC-aCSF-NMDA or aCSF-BICC-leptin-NMDA time course images. All individual neurons expressing GCaMP6 were manually outlined and saved as a ROI profile. The GCaMP6-expressing neuron ROI was applied to the full time lapse recording at 488 emission (Int488) using Time Measurement function. Viable neurons were defined as those in which the increase in Int488 in response to NMDA was greater than 2.5% (in all cases, the NMDA response was greater than 2 SD of the initial baseline activity) and were selected for further analysis. The relative change in Int488 was calculated for each viable neuron using $(F- F_0)/F_0$ where $F_0$ represents the average Int488 over the initial 1 min of the aCSF-BICC or the aCSF-BICC-leptin/aCSF-BICC-aCSF 90 min perfusion and F is the average Int488 during final 1 min of the same test perfusion. To determine if neuronal calcium activity increased or decreased, the average of the first 5 min of normalized aCSF-BICC-aCSF or aCSF-BICC-leptin recording was compared to the final 5 min.

## Statistical analysis

Data are expressed as Mean ± SEM. The physiological experiments in *Figures 1*, *2*, *3*, *4*, *5*, *9*, *13* and *18* were analyzed using two-way repeated measures ANOVA and the post-hoc Newman Keuls test. In *Figure 8* (ISH), responses to PVN leptin were assessed using one-way

repeated measures ANOVA and the post-hoc Newman Keuls test; differences in the number of ISH grains at each PVN level were determined using a paired-t test.

## Acknowledgements

The authors are grateful for the technical assistance of Peter Levasseur, Dr. Brian Jenkins, and for the assistance provided by Dr. Stefanie Kaech-Petrie (and the Advance Light Microscopic Core, funded in part by P30 NS061800—PI Dr. Sue Aicher) and Dr. Todd Stincic as we developed the GCaMP-6 protocol.

## Additional information

### Funding

| Funder | Grant reference number | Author |
|---|---|---|
| National Institutes of Health | HL088552 | Virginia L Brooks |
| National Institutes of Health | HL128181 | Virginia L Brooks |
| National Institutes of Health | CA217989 | Daniel L Marks |
| National Institutes of Health | NS099503 | Andrei D Sdrulla |
| National Institutes of Health | DK112198 | Christopher J Madden |

The funders had no role in study design, data collection and interpretation, or the decision to submit the work for publication.

### Author contributions

Zhigang Shi, Jennifer Wong, Conceptualization, Data curation, Formal analysis, Validation, Investigation, Visualization, Methodology, Writing - review and editing; Nicole E Pelletier, Conceptualization, Resources, Data curation, Formal analysis, Validation, Investigation, Visualization, Methodology, Writing - review and editing; Baoxin Li, Formal analysis, Validation, Investigation; Andrei D Sdrulla, Conceptualization, Formal analysis, Validation, Methodology, Writing - review and editing; Christopher J Madden, Conceptualization, Methodology, Writing - review and editing; Daniel L Marks, Conceptualization, Funding acquisition, Methodology, Project administration, Writing - review and editing; Virginia L Brooks, Conceptualization, Resources, Data curation, Formal analysis, Supervision, Funding acquisition, Validation, Investigation, Methodology, Writing - original draft, Project administration, Writing - review and editing

### Author ORCIDs

Zhigang Shi https://orcid.org/0000-0002-5828-1904
Daniel L Marks http://orcid.org/0000-0003-2675-7047
Virginia L Brooks https://orcid.org/0000-0001-6709-6631

### Ethics

Animal experimentation: This study was performed in strict accordance with the recommendations in the Guide for the Care and Use of Laboratory Animals of the National Institutes of Health. All of the animals were handled according to approved institutional animal care and use committee (IACUC) protocols (TR01_IP00000151) of Oregon Health & Science University. All surgery was performed under isoflurane, alpha-chloralose, or pentobarbital anesthesia, and every effort was made to minimize suffering.

### Decision letter and Author response

Decision letter https://doi.org/10.7554/eLife.55357.sa1
Author response https://doi.org/10.7554/eLife.55357.sa2

## Additional files

### Supplementary files
• Transparent reporting form

### Data availability
All data generated and analyzed are included in the manuscript. Source data files are provided for relevant figures.

The following dataset was generated:

| Author(s) | Year | Dataset title | Dataset URL | Database and Identifier |
|---|---|---|---|---|
| Shi Z, Pelletier NE, Wong J, Li B, Sdrulla AD, Madden CJ, Marks DL, Brooks VL | 2020 | Leptin increases sympathetic nerve activity via induction of its own receptor in glutamatergic neurons in the paraventricular nucleus | http://dx.doi.org/10.5061/dryad.wwpzgmsg5 | Dryad Digital Repository, 10.5061/dryad.wwpzgmsg5 |

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
