## [Decision Letter]

**Acceptance summary:**

The reviewers were in agreement that your multi-disciplinary approaches provide novel findings about the role of obesity, increased leptin and blood pressure will be of wide interest. You also responded to the previous critiques in a very thorough manner and have tempered your conclusions appropriately.

**Decision letter after peer review:**

Thank you for submitting your article "Leptin increases sympathetic nerve activity via induction of its own receptor in the paraventricular nucleus" for consideration by *eLife*. Your article has been reviewed by two peer reviewers, one of whom is a member of our Board of Reviewing Editors, and the evaluation has been overseen by Catherine Dulac as the Senior Editor The following individual involved in review of your submission has agreed to reveal their identity: Malcolm Low (Reviewer #2).

The reviewers have discussed the reviews with one another and the Reviewing Editor has drafted this decision to help you prepare a revised submission.

Summary:

Shi and colleagues describe results from a series of studies investigating the role of leptin action in the paraventricular nucleus of the hypothalamus (PVN) in regulating sympathetic nerve activity. They also provide evidence that leptin activates PVN neurons and leptin receptors which are lowly expressed in the PVN are upregulated in the context of obesity. This could contribute to comorbidities including hypertension in the obese state. The manuscript is well written, clear and concise. The authors use both rats and mice (and both sexes) with a combination of physiological, pharmacological and neuroanatomic approaches to investigate in detail the importance of leptin receptors expressed in the PVN to excite presympathetic neurons and thereby increase lumbar SNA and BAT SNA, with consequent elevations of BP, HR and BAT thermogenesis. Their overall approach was comprehensive and rigorous and addressed a novel hypothesis that leptin can increase the expression of its own receptor in the PVN. After validating the proposed upregulation of LepR by its agonist, the authors go on to show that virtually all the cells in the PVN with LepR are glutamatergic neurons, a proportion of these are TRH expressing and some of the triple-positive neurons project to the RVLM based on the retrograde transport of CTB from RVLM to PVN. Like the detailed analyses of peripheral SNA, the FISH data on LepR expression in PVN is convincing. Collectively, the studies are well done and provide novel information that will be of interest to the field.

Essential revisions:

Did the authors map the respective injection sites? If so, these could be included in a composite anatomic cartoon.

The authors should directly discuss potential differences between mice and rats as there are likely species differences in the expression of leptin receptors and organization of the PVN.

The panels in some of the figures are small. Maybe it would make sense to break them up into bigger format and more figures.

Each panel in the figures needs a label.

The photomicrographs are too dark. They could be enhanced to highlight the label (or lack of it).

Unfortunately, Figure 7 was missing from both the combined manuscript files and the individual files so I cannot comment of that data specifically. Finally, the authors used calcium imaging in PVN slices to study the activation of unidentified glutamatergic neurons by leptin, with the striking finding that leptin had mixed effects (more cells showing increased activity while fewer showed decreased activity) but in all cases the regular bursting activity was blocked. Their data is nicely summarized in Figure 15 to provide a physiological context for the relative importance of the PVN and ArcN on SNA and the HPT axis under conditions of increasing leptin vs. decreasing leptin. Although not a main thrust of the paper, their FISH analysis of LepR expression confirmed that in vivo there is almost no expression of the mRNA in either microglia or astrocytes, in contrast to many reports of LepR expression in glia under in vitro conditions.

Although slice electrophysiology to refine the direct actions of leptin on PVN glutamatergic neurons might better define the specific ionic currents, membrane potential changes and action potential generation compared to the GCamP-6 calcium imaging approach, I don't believe it would make a difference to the main conclusions of the paper.

The lag time to measure increased LepR expression in the PVN in response to leptin appears to be considerably longer than the time frame for SNA activation by PVN nanodosing with leptin. It would be worth some more discussion of this point in light of the fact that leptin activates presympathetic neurons in the medulla by several distinct neural circuits and the complex interplay of these circuits ultimately activates peripheral SNA. Similarly, some comment on the relevance of both fast synaptic release of glutamate and presumably delayed TRH release from the same PVN projections to medullary nuclei would strengthen an already solid paper relevant to scientists in the areas of both metabolism and cardiovascular physiology.

---

## [Author Response]

Essential revisions:Did the authors map the respective injection sites? If so, these could be included in a composite anatomic cartoon.

Yes, these sites are shown individually in Figure 20. It would be difficult to combine these individual sites in a composite, given the large number of protocols employed and the extensive overlap of injection sites.

The authors should directly discuss potential differences between mice and rats as there are likely species differences in the expression of leptin receptors and organization of the PVN.

This is an important point, which we now address in a new Perspectives section.

The panels in some of the figures are small. Maybe it would make sense to break them up into bigger format and more figures.

Yes, done for some of the figures.

Each panel in the figures needs a label.

We have added labels to figures summarizing experimental data. We also added letters to many of the different panels in figures illustrating photomicroscopic data, but in others each panel is instead identified by other features such as hypothalamic site and mm from bregma (when a series of sections are shown).

The photomicrographs are too dark. They could be enhanced to highlight the label (or lack of it).

Because LepR expression and the puncta representing the mRNA expression are low in the PVN, we used several new strategies to enhance its visibility. First, we changed the red puncta to magenta, which contrasts better against the largely black background. Second, because we did not quantify the signal, we adjusted the gamma (in the histogram), as well as the brightness and contrast, in some figures. This additional adjustment has been added to the Materials and methods. Third, as suggested, we split many figures into two to enlarge panels. Finally, in some figures, the LepR signal is still difficult to see with no magnification, but is easily seen when one zooms in on the displayed figure. Therefore, we added a comment in some figure legends that to optimally view LepR puncta, the image(s) should be surveyed online using the zoom feature.

Unfortunately, Figure 7 was missing from both the combined manuscript files and the individual files so I cannot comment of that data specifically.

We apologize for this oversight. This figure is now included (now Figure 8).

Finally, the authors used calcium imaging in PVN slices to study the activation of unidentified glutamatergic neurons by leptin, with the striking finding that leptin had mixed effects (more cells showing increased activity while fewer showed decreased activity) but in all cases the regular bursting activity was blocked. Their data is nicely summarized in Figure 15 to provide a physiological context for the relative importance of the PVN and ArcN on SNA and the HPT axis under conditions of increasing leptin vs. decreasing leptin. Although not a main thrust of the paper, their FISH analysis of LepR expression confirmed that in vivo there is almost no expression of the mRNA in either microglia or astrocytes, in contrast to many reports of LepR expression in glia under in vitro conditions.

Yes, thank you.

Although slice electrophysiology to refine the direct actions of leptin on PVN glutamatergic neurons might better define the specific ionic currents, membrane potential changes and action potential generation compared to the GCamP-6 calcium imaging approach, I don't believe it would make a difference to the main conclusions of the paper.

One of the challenges in assessing the cellular effects of leptin using electrophysiology was that the cellular events are slowly developing. The use of GCaMP-6 allowed us to capture the slow excitation (in a subset of the neurons) over a 90 min period.

The lag time to measure increased LepR expression in the PVN in response to leptin appears to be considerably longer than the time frame for SNA activation by PVN nanodosing with leptin. It would be worth some more discussion of this point in light of the fact that leptin activates presympathetic neurons in the medulla by several distinct neural circuits and the complex interplay of these circuits ultimately activates peripheral SNA. Similarly, some comment on the relevance of both fast synaptic release of glutamate and presumably delayed TRH release from the same PVN projections to medullary nuclei would strengthen an already solid paper relevant to scientists in the areas of both metabolism and cardiovascular physiology.

Again, a good suggestion. We have added a new paragraph in which we speculate on additional mechanisms by which the effects of leptin can be amplified over time, including as you suggest release of TRH and an interplay of downstream circuits.